# DISENTANGLING INSTRUCTION INFLUENCE IN DIFFUSION TRANSFORMERS FOR PARALLEL MULTI-INSTRUCTION-GUIDED IMAGE EDITING

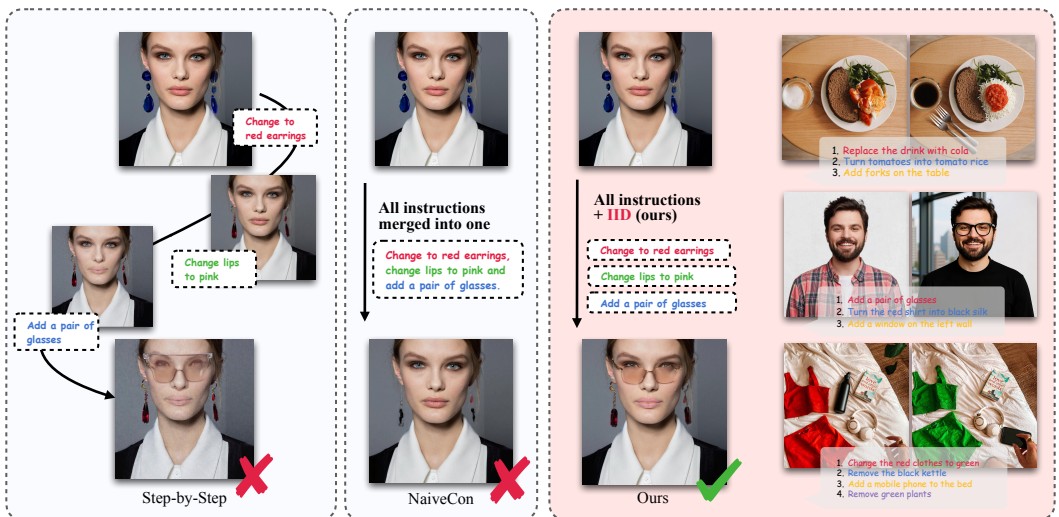

Figure 1: Multi-instruction guided image editing is challenging because step-by-step editing, using the result from one step as input for the next, often leads to gradual quality loss and accumulated distortions. Meanwhile, NaiveCon, which simply concatenates all instructions into a composite one, frequently fails to accomplish each desired change. In contrast, our proposed method can complete all instructions in one step with negligible degradation.

## ABSTRACT

Instruction-guided image editing allows users to specify modifications in natural language, providing greater flexibility and control. Recent advances show that Diffusion Transformers (DiTs) surpass U-Net-based diffusion models in both scalability and performance. However, real-world scenarios often involve applying multiple instructions simultaneously. Step-by-step editing accumulates errors and degrades image quality, while putting all instructions into a single prompt typically leads to incomplete edits. To address these challenges, we introduce Instruction Influence Disentanglement (IID), a framework that enables simultaneous handling of multiple instructions within a single denoising process for DiT-based models. By analyzing the self-attention mechanisms of DiTs, we observe distinct attention patterns under multi-instruction settings and design instruction-specific masks to disentangle their influence. These masks guide the editing process to achieve localized modifications while preserving consistency in non-edited regions. Extensive experiments demonstrate that IID improves both fidelity and instruction completion, while also reducing computational overhead compared to existing approaches.

## 1 INTRODUCTION

Instruction-guided image editing (Brooks et al., 2023; Gu et al., 2024; Sheynin et al., 2024; Ma et al., 2024; Zhang et al., 2023; Liu et al., 2025) has gained increasing attention for its ability to enable users to specify editing objectives using natural language. Among various frameworks, Diffusion

Transformers (DiTs) (Peebles & Xie, 2023; Esser et al., 2024) exhibit superior scalability, with performance improving as model size and the number of training data increase (Xiao et al., 2024; Paul, 2024; Batifol et al., 2025), surpassing U-Net-based diffusion models (Brooks et al., 2023; Huang et al., 2025; Zhang et al., 2024; Hui et al., 2025). However, these methods primarily perform well in single-instruction editing, while real-world scenarios often require multiple modifications to concurrently process an image (Guo & Lin, 2024; Batifol et al., 2025; Ma et al., 2024; Khodadadeh et al., 2022). Given DiTs' excellent performance in single-instruction-guided image editing, extending them to multi-instruction editing is a natural yet challenging research direction.

Although applying an editing model step by step to each instruction (Zhang et al., 2023; Ma et al., 2024; Joseph et al., 2024) or merging multiple instructions into a composite one may appear feasible, both approaches have significant limitations. As shown in Fig. 1, the sequential editing often leads to progressive distortions and degradation (e.g., the blurred glasses and facial features) as the number of edits increases. This deterioration likely stems from cumulative errors introduced by repeated denoising processes, which disrupt the natural diffusion feature space (Lin et al., 2024; Sheynin et al., 2024). Similarly, the naive instruction concatenation often fails to handle all intended modifications, typically applying only one successfully. This issue may arise from the instructions' dual role in enforcing edits on target regions while preserving the unedited areas. When multiple instructions are processed simultaneously, they tend to compete for dominance, and stronger ones may dominate weaker ones, leading to incomplete editing, a phenomenon we term instruction conflict. Additionally, the theoretical analysis of error accumulation and instruction conflicts is provided in Appendix A.4.

Intuitively, disentangling the influence of multiple instructions to ensure each one only affects its target region can mitigate such conflicts, thus enabling parallel image editing with various instructions in a single pass. However, prior work (Guo & Lin, 2024; Huang et al., 2025; Xie et al., 2023; Hertz et al., 2022) has focused on U-Net-based diffusion models to leverage cross-attention maps between textual edited object tokens and noised images to generate spatial masks to guide attention computations. These techniques perform suboptimally in DiTs due to fundamental architectural differences, particularly the replacement of U-Net (Ronneberger et al., 2015) with multi-head self-attention transformer blocks (Vaswani et al., 2017), and require prior identification of editable text objects. Other approaches (Couairon et al., 2022; Song et al., 2025) depend on external image segmenters to generate instruction-specific masks, which becomes increasingly complex and less scalable in multi-instruction scenarios.

To address this gap, we analyze the self-attention mechanism in DiTs by visualizing attention maps between instruction tokens and noisy image tokens, as well as among image tokens themselves in Sec . 4.1. We employ Omnigen (Xiao et al., 2024) and FluxEdit (Paul, 2024), which represent diverse DiT architectures. Our findings show that, after several steps of reverse diffusion, the semantics of an instruction, approximated by the averaged attention of all text tokens, tend to focus on the edited regions, which reduces the reliance on explicit object tokens extraction or image segmenter for instruction influence localization as required by U-Net-based models. Moreover, different instructions often yield similar head-wise attention patterns for a given image, like highlighting their respective editing regions or maintaining consistent global attention.

Based on these observations, we propose **I**nstruction **I**nfluence **D**isentanglement (**IID**), a training-free framework for parallel multi-instruction image editing in a single pass, tailored to various DiT-based frameworks. Concretely, during the early reverse diffusion phase, individual instructions are processed independently. At a predefined timestep, attention maps between instruction tokens and noised image tokens are extracted from a designated model layer. To disentangle instruction influence, we apply a head-wise mask generation strategy, where for each attention head, we subtract the mean attention map of all other instructions from the current instruction and then aggregate the results across heads to form the final mask. This subtraction can effectively mitigate inter-instruction interference and suppress influence over non-editing regions. Instructions are then adaptively concatenated based on their influence scores, estimated via average attention weights over editing areas, and their latent image representations are blended according to the computed masks. A new attention mask is constructed for the composed instruction-latent image pair, and the diffusion process finally resumes to generate the final image. Quantitative and qualitative evaluations on the MagicBrush dataset (Zhang et al., 2023) and a custom-collected dataset demonstrate that IID can achieve superior performance in both fidelity and instruction completion, validating its effectiveness for parallelized multi-instruction guided image editing.

The contribution of our work is summarized as follows: 1) We conduct an in-depth investigation of self-attention mechanisms in DiTs for instruction-guided image editing, uncovering unexplored insights to inform future research. 2) We propose a novel framework that enables the parallel processing of multiple edits in a single denoising process. It not only significantly reduces diffusion steps but also improves editing performance, including lower noise generation and better consistency in non-edited regions compared to step-by-step editing. 3) Our framework is extensively evaluated on open-source and custom-constructed multi-instruction editing datasets, demonstrating versatility on Omnigen, FluxEdit, and the recent state-of-the-art FluxKontext (Batifol et al., 2025).

## 2 RELATED WORKS

**Text-guided editing via Diffusion Models.** Previous diffusion-based editing methods (Hertz et al., 2022; Kulikov et al., 2024; Cao et al., 2023; Mokady et al., 2023; Xu et al., 2023) are built upon text-to-image models and require the caption of the target image and source image as inputs. These approaches employ inversion-based techniques (Song et al., 2021; Meng et al., 2022; Avrahami et al., 2024), where the initial noise corresponding to the source image is extracted and then denoised to generate the edited image. During this process, the denoising trajectory of the target image can be refined through attention control (Hertz et al., 2022; Cao et al., 2023), optimization techniques (Mokady et al., 2023; Kawar et al., 2023), and user-provided masks (Xie et al., 2023; Avrahami et al., 2023). Recently, instruction-guided image editing methods (Zhang et al., 2024; Geng et al., 2024; Huang et al., 2024; Wei et al., 2024; Xiao et al., 2024; Shi et al., 2024; Brooks et al., 2023; Hui et al., 2025) have attracted increasing research interest, as they provide a more user-friendly experience without the need for image captions. These approaches fine-tune pretrained text-to-image diffusion models using a conditional image generation objective. Moreover, due to the powerful capabilities of DiTs (Esser et al., 2024; Peebles & Xie, 2023), recent state-of-the-art editing models (Wei et al., 2024; Shi et al., 2024; Liu et al., 2025) (e.g., FluxEdit (Paul, 2024), Omnigen (Xiao et al., 2024) and FluxKontext (Batifol et al., 2025)) have shifted their backbone from U-Net to DiT architectures.

**Multi-instruction Guided image editing.** While instruction-guided image editing models perform well in single-instruction scenarios, they struggle with multiple instructions. Simply merging instructions into a single prompt fails to address this limitation due to the instruction conflict, where models often prioritize one instruction over others. While FOI (Guo & Lin, 2024) leverages cross-attention of U-Net (Ronneberger et al., 2015) to localize edits and employs a modulation module to isolate target regions, this approach is incompatible with DiT-based architectures due to unresolved challenges in effective mask extraction and the neglect of self-attention of visual and textual tokens. Moreover, FOI processes all instructions simultaneously, whereas our method handles composite instructions after a predefined step, thus reducing computational overhead. Beyond these, prior works (Huang et al., 2025; Bar-Tal et al., 2023) integrate multiple diffusion processes via optimization techniques or attention control to address multi-condition image generation in inversion-based editing. However, these methods require task-specific designs and are unsuitable for instruction-guided editing.

## 3 PRELIMINARY

Instruction-guided image editing aims to transform a source image $I_v$ into a target image $I_g$ according to a textual instruction $P$. This section provides an overview of the fundamental concepts of Diffusion Models (DMs) for image editing and self-attention mechanisms in Diffusion Transformers (DiTs).

**Diffusion Model for Image Editing.** DMs (Song et al., 2020; Ho et al., 2020) operate through a *forward diffusion process* that progressively adds noise to data and a *reverse diffusion process* that aims to reconstruct the original data from noise through iterative denoising of the Gaussian noise. Following Latent Diffusion Models (Rombach et al., 2022), the source image $I_v$, target image $I_g$ and text prompt $P$ are encoded into latent representations as $c_I$, $z_0$, and $c_P$ by corresponding encoders. To improve training stability and sampling efficiency, recent DiT-based models adopt the Rectified Flow framework (Liu et al., 2022), which replaces the traditional denoising process with a continuous-time ODE formulation. Instead of learning to remove noise step-by-step, the model learns a time-dependent vector field $\mathbf{v}_\theta(z_t, t, c_I, c_P)$ that transports noisy latent $z_t$ toward target image $z_0$, along deterministic paths, denoted as

$$\frac{dz}{dt} = \mathbf{v}_\theta(z_t, t, c_I, c_P), \tag{1}$$

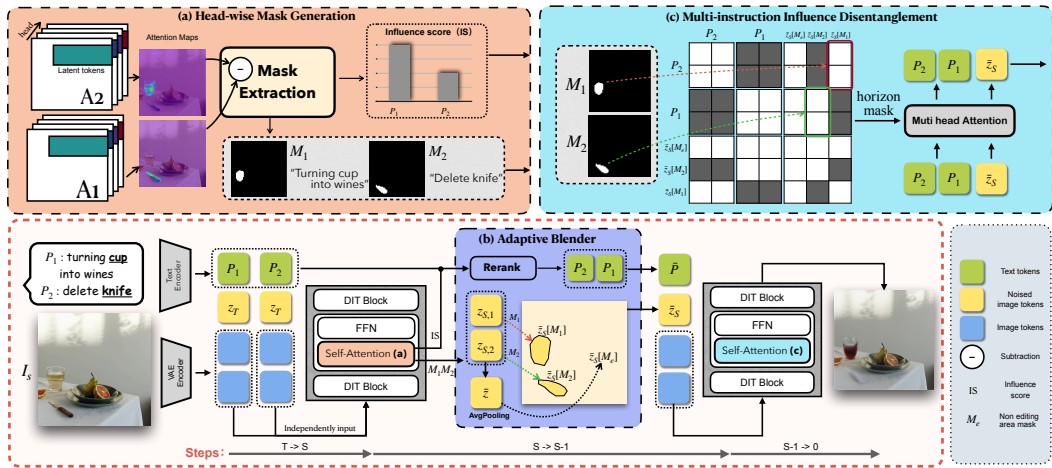

Figure 2: Illustration of our IID framework. (a) The editing region for each instruction is localized using a head-wise mask generation strategy at a predefined step $S$; (b) All instructions are concatenated based on their influence scores, and their latent representations are blended based on their masks, producing the compositional instruction and latent images; (c) A new attention mask is constructed to disentangle the influences of various instructions to guide the subsequent denoising.

where $z_1 \sim \mathcal{N}(0, 1)$ is the initial noisy latent, and $z_t = (1 - t)z_0 + tz_1$ represents linear interpolation between $z_0$ and $z_1$. Finally, the model is trained to predict the true displacement velocity at each step.

**Self-Attention in Diffusion Transformers.** DiTs decompose the noisy target image $z_t$ into a sequence of $N_z$ patch tokens, denoted as $\{z_{t,i}\}_{i=1}^{N_z}$, where $t$ represents the timestep. For the conditional information, the text instruction $P$ is tokenized into $N_p$ text tokens and encoded as $\{p_i\}_{i=1}^{N_p}$, while the source image $I_v$ is similarly decomposed into $N_v$ patch tokens $\{v_i\}_{i=1}^{N_v}$. These tokens are processed through multiple stacked multi-head self-attention layers to predict the noise velocity at each timestep. The fundamental self-attention mechanism is formulated as:

$$A^j(Q^j, K^j, V^j) = \text{softmax}\left(\frac{Q^j K^{j\top}}{\sqrt{d}}\right) V^j \tag{2}$$

While various DiT architectures employ different approaches to incorporate conditional information, Peebles & Xie (2023) categorizes them into two primary classes, including adaLN-Zero (e.g., FluxEdit (Paul, 2024)) and In-Context Conditioning (e.g., Omnigen (Xiao et al., 2024) and FluxKontext (Batifol et al., 2025)). Specifically, FluxEdit concatenates text tokens and noisy image tokens as input to compute the $Q^j$, $K^j$, and $V^j$ matrices, while image conditioning $I_v$ is incorporated into the noisy image before token generation. In contrast, the other models concatenate all conditional tokens and noisy image tokens to compute the key and value matrices.

## 4 METHODOLOGY

In multi-instruction-guided image editing, we are given a set of text instructions $\{P_1, \ldots, P_N\}$, where $N$ denotes the number of instructions. Given a source image $I_v$, the objective is to apply all instructions to generate the target image $I_g$ with height $H$ and width $W$. However, when multiple instructions are concatenated into a single one, the model often results in instruction conflicts, prioritizing some edits while neglecting others. To address this, as illustrated in Fig. 2, we use a head-wise mask generation strategy to generate a mask for each instruction. The text and noisy image tokens are then adaptively blended to form compositional instructions and latent images, which are finally taken as input by the editing model to denoise over subsequent timesteps. This section analyzes the attention weight in Sec. 4.1 and details the IID framework in Sec. 4.2.

Figure 3: The visualization of the attention map between instruction tokens and noise image tokens $\bar{A}_{ZP}$ and among noise image tokens $\bar{A}_{ZZ}$ and the pipeline of our Head-wise Mask Generation strategy. Attention weights are extracted from the penultimate layer. "Avg" represents the averaging attention map across all heads.

## 4.1 ATTENTION WEIGHT ANALYSIS

To mitigate instruction conflicts, it is crucial to ensure that each instruction $P_i$ does not influence regions edited by other instructions through masks. While prior work requires separate models to identify the textual tokens corresponding to the editable object and is limited to U-Net-based architectures, we first analyze the attention weight $A^j$ of the self-attention in DiTs.

Given that FluxEdit and OmniGen employ distinct token concatenation strategies to construct the query, key, and value matrices, resulting in different attention weight distributions, we analyze attention patterns shared by both models to provide a generalized understanding on DiT architectures. Specially, for the $j$-th attention head, we analyze the attention weights between the the noisy image token sequence $\{z_i\}_{i=1}^{N_z}$ and instruction token sequence $\{p_i\}_{i=1}^{N_p}$, represented by $A_{ZP}^j \in \mathbb{R}^{N_z \times N_p}$ and the attention weights among the noisy image tokens, denoted as $A_{ZZ}^j \in \mathbb{R}^{N_z \times N_z}$. Then $A_{ZP}^j$ and $A_{ZZ}^j$ are averaged along the second dimension to capture the semantics of the entire instruction, yielding an $N_z$-dimensional vector. The vector is subsequently min-max normalized and reshaped into attention maps as $\bar{A}_{ZP}^j$ and $\bar{A}_{ZZ}^j$, each with a resolution of $H' \times W'$ where $H' = H//q$, $W' = W//q$ and $q = 16$.

Finally, we visualize both types of attention maps extracted from FluxEdit and Omnigen in Fig. 3. The results reveal several key insights: 1) The attention maps of DiTs exhibit diverse patterns, such as emphasizing edited regions, focusing on unedited areas, or prioritizing overall image reconstruction, suggesting that traditional head averaging is suboptimal for locating editing regions in DiTs. We speculate that it is possible because self-attention among tokens propagates information holistically rather than relying on individual tokens. 2) Despite architectural differences between models, when provided with the same input image $I_v$, many attention heads demonstrate similar functional behavior across different instructions. 3) The attention maps $\bar{A}_{ZP}^j$ and $\bar{A}_{ZZ}^j$ both exhibit the same two phenomena described above. For a more detailed discussion, please refer to the Appendix. A.5.

## 4.2 INSTRUCTION INFLUENCE DISENTANGLEMENT FRAMEWORK

**Head-wise Mask Generation.** While previous analyses have shown that for each instruction, most attention heads can attend to its corresponding editing region, accurately isolating these regions remains challenging because numerous other attention heads maintain a different focus, reducing the contrast between edited and non-edited areas in the averaged attention map (i.e., "Avg" in Fig. 3). Additionally, even in heads that primarily attend to the editing region, high-intensity noise persists in non-editing areas (e.g., head 14 in Fig. 3 (c) and head 7 in Fig. 3 (b)), which is difficult to eliminate through post-processing methods such as threshold-based filtering.

Fortunately, for the same source image, we observe that different instructions can exhibit similar attention patterns in the same heads. This finding motivates us to design a head-wise mask generation strategy for extracting editing masks. Let's take the attention map $\bar{A}_{ZP}^j$ between instruction tokens $\{p_i\}_{i=1}^{N_p}$ and noisy image tokens $\{z_i\}_{i=1}^{N_z}$, as an example. For a given instruction $P_i$, we first compute

the attention map difference by subtracting the averaging attention map of the same head across all other instructions. The negative values are then set to zero to suppress non-relevant regions. This process can be expressed as follows:

$$M_i^j = \max(0, \bar{A}_{ZP_i}^j - \frac{1}{N-1}\sum_{k \neq i}^N \bar{A}_{ZP_k}^j), \tag{3}$$

where $M_i^j \in \mathbb{R}^{H' \times W'}$ is the editing region-focused attention map for $P_i$ in the $j$-th head. Such a head-wise subtraction effectively reduces the attention weights in non-editing regions of $M_i^j$, because the attention weights of the same head in these areas are similar across different instructions, leading to near-zero values in $M_i^j$. Moreover, the subtraction also causes the weights corresponding to the editing regions of other instructions to become negative, which are then suppressed to zero in $M_i^j$.

Next, to obtain the final mask $M_i \in \mathbb{R}^{H' \times W'}$ for $P_i$, we average $M_i^j$ across different heads, then apply a Gaussian Filter to smooth the results following (Hertz et al., 2022), and perform binarization using Otsu's Filter (Otsu, 1979), which automatically determines the threshold without manual intervention. Let $J$ denote the number of attention heads. The process can be summarized as follows:

$$M_i = \text{Otsu's Filter}(\text{Gaussian Filter}(\sum_{j=1}^J M_i^j/J)). \tag{4}$$

**Adaptive Blender.** To enable parallel instruction processing and reduce computation, we concatenate all instructions into a compositional form $\bar{P}$ and aggregate the corresponding noised images $z_{S,i}$ into $\bar{z}_S$ at timestep $S$. However, the different instruction ordering can often introduce bias, as earlier instructions may dominate, exacerbating instruction conflicts where weaker instructions fail to complete. To mitigate this issue, for OmniGen and FluxKontext, we ensure all $P_i$ share the same position embedding by padding each instruction to a uniform length and assigning the same position id sequence during the encoding stage, thus neutralizing positional bias. For FluxEdit, which lacks position embeddings, we compute $\sum j = 1^J (\bar{A}_{ZP}^j \cdot M_i)$ followed by normalization as the Influence Score (IS) of each instruction on its corresponding editing region. The instructions are then sorted in ascending order based on these scores to approximately equalize their editing influence.

Next, to construct the compositional noisy image $\bar{z}_S$, we compute the averaging latent representation of all instructions at timestep $S$, denoted as $\bar{z}_{S,0} = \sum_{i=1}^N z_{S,i}/N$. Then, for each instruction, we update the corresponding masked regions in this averaged representation using the respective $z_{S,i}$:

$$\bar{z}_{S,i} = z_{S,i} \odot M_i + \bar{z}_{S,i-1} \odot (1 - M_i), \tag{5}$$

where $i \in [1, N]$ and the compositional noisy image $\bar{z}_S = \bar{z}_{S,N}$. For overlapping mask regions across multiple instructions, these regions are replaced with the averaged values from the corresponding $z_{S,i}$ to consider the information of all instructions.

**Multi-instruction Influence Disentanglement.** To disentangle the influence of each instruction to make each one does not interfere with the others' editing regions, we construct an attention mask between the tokens of the compositional instruction $\bar{P}$ and the noised image $\bar{z}_S$. While instruction tokens attend to all tokens of the noisy image before timestep $S$, we modify this strategy so that the token of $P_i$ can only attend to noisy image tokens excluding regions masked by $M_j$, where $j \in [1, N]$ and $j \neq i$. This is achieved by setting the attention scores between the token of $P_i$ and the excluded image regions to $-\infty$. Especially, in the case of overlapping regions between $M_i$ and other masks, the $P_i$ tokens are still allowed to attend to the corresponding image tokens. Finally, using the newly constructed attention mask, we replace $z_t$ and $C_P$ in Eq. 1 with blended $\bar{z}_S$ and $\bar{P}$, respectively, and perform denoising over subsequent timesteps to ensure that the generated image accurately reflects all intended edits.

## 5 EXPERIMENTS

### 5.1 EXPERIMENT SETTING

**Benchmark.** Due to the limited availability of multi-instruction editing benchmarks, we primarily evaluate on the test split of the MagicBrush dataset (Zhang et al., 2023), which includes

Figure 4: Qualitative comparison results of our proposed IID against Step-by-Step and NaiveCon. Each row from top to bottom corresponds to FluxEdit, Omnigen, and FluxKontext, respectively.

sessions with two or three instructions, as well as the recently released Kontext-Bench (Batifol et al., 2025). To increase evaluation complexity and diversity, we further curate a new dataset with up to six instructions, using images from public datasets and realistic applications.

**Metrics.** Following Zhang et al. (2023), we utilize L1 and L2 to quantify pixel-level differences between generated and ground truth images, CLIP-I and DINO to measure the image quality with the cosine similarity between the generated image and ground truth image using their CLIP (Radford et al., 2021) and DINO (Caron et al., 2021) embeddings, and CLIP-T (Ruiz et al., 2023) to measure the text-image alignment using the cosine similarity between the descriptions of ground truth images and the CLIP embeddings of generated images. Since our own dataset does not contain ground truth, we report these quantitative metrics on the MagicBrush dataset. Following Guo & Lin (2024), We conduct a human preference study with forty participants to evaluate editing quality in terms of instruction alignment and image fidelity across three datasets. For instruction alignment, participants are asked to select the method that best reflects the intent of all given instructions. For image alignment, they are instructed to choose the method that best preserves the visual details of the original image.

Table 1: Quantitative Results on MagicBrush Dataset. The best results are highlighted in **bold**.

| Method | L1↓ | L2↓ | CLIP-I↑ | DINO↑ | CLIP-T↑ |
|---|---|---|---|---|---|
| *Target caption-guided* | | | | | |
| SD-SDEdit | 0.2090 | 0.0813 | 0.7467 | 0.5435 | 0.3241 |
| NTI | 0.2120 | 0.0846 | 0.6928 | 0.4117 | 0.3095 |
| ParaEdit | 0.2943 | 0.1438 | 0.6975 | 0.3571 | 0.3108 |
| *Instruction-guided* | | | | | |
| HIVE | 0.1521 | 0.0894 | 0.7108 | 0.4551 | 0.2684 |
| InP2P | 0.2526 | 0.1109 | 0.6675 | 0.3717 | 0.2646 |
| HQ-EDIT | 0.3040 | 0.1480 | 0.6656 | 0.4216 | 0.2676 |
| InP2P-Mgic | 0.1463 | 0.0598 | 0.8258 | 0.6952 | 0.3095 |
| SmartEdit | 0.1574 | 0.0567 | 0.7248 | 0.5384 | 0.2784 |
| *Omnigen* | | | | | |
| Step | 0.1617 | 0.0703 | 0.8337 | 0.7115 | 0.2699 |
| NaiveCon | 0.1394 | 0.0525 | 0.8437 | 0.7353 | 0.2754 |
| FOI | 0.1223 | 0.0523 | 0.8363 | 0.7312 | 0.2643 |
| **IID** | 0.1128 | 0.0466 | 0.8599 | 0.7604 | 0.2805 |
| *FluxEdit* | | | | | |
| Step | 0.1339 | 0.0457 | 0.8123 | 0.6586 | 0.2645 |
| NaiveCon | 0.1221 | 0.0455 | 0.8317 | 0.7140 | 0.2623 |
| **IID** | 0.1112 | 0.0403 | 0.8445 | 0.7222 | 0.2778 |
| *Flux-Kontext-dev* | | | | | |
| Step | 0.1131 | 0.0423 | 0.8704 | 0.7725 | 0.3645 |
| NaiveCon | 0.1016 | 0.0408 | 0.8689 | 0.7690 | 0.3423 |
| **IID** | **0.0914** | **0.0394** | **0.8723** | **0.7734** | **0.3668** |

**Baselines.** Given the lack of multi-instruction-guided image editing methods for DiT-based architectures, we benchmark our proposed framework, IID, against two strategies: (1) step-by-step editing (Step) and (2) naive concatenation of multiple instructions (NaiveCon). For DiT-based instruction-guided editing models, we include FluxEdit (Paul, 2024), Omnigen (Xiao et al., 2024), and FluxKontext (Batifol et al., 2025), which represent the mainstream DiT architecture introduced by (Peebles & Xie, 2023). Additionally, we compare against target caption-guided editing approaches such as SD-SDEdit (Shi et al., 2024), Null Text Inversion (NTI) (Mokady et al., 2023), and the multi-aspect parallel editing method ParaEdit (Huang et al., 2025). We further consider U-Net-based instruction-guided editing models, including HIVE (Zhang et al., 2024), InstructPix2Pix (InP2P) (Brooks et al., 2023), HQ-EDIT (Hui et al., 2025), InP2P-Magic (Zhang et al., 2023), and SmartEdit (Huang et al., 2024). While FoI is originally designed for U-Net-based diffusion models, we adapt it to OmniGen for comparison.

**Implementation details.** For all three models, we select the 10% step in the total diffusion process for mask generation and blending ($S = 0.9T$). Unless otherwise specified, we extract the editing mask from $A_{ZP}$ in the penultimate layer of both models.

Due to page limitations, further details on the experiment setting can be found in the Appendix A.1.

Figure 5: Qualitative comparison results of instruction-guided DiT-based models enhanced by our proposed IID against NaiveCon using other U-Net-based editing models as the backbone model.

## 5.2 MAIN RESULTS

**Quantitative evaluation.** As shown in Table 1, the DiT-based models consistently outperform U-Net-based models across nearly all evaluation metrics. This observation underscores the superiority of DiT architectures for image editing tasks. Furthermore, our proposed IID significantly improves performance over the Step and NaiveCon on all three DiT-based models, with notable reductions in L1/L2 errors and increases in CLIP-I, DINO, and CLIP-T scores. These improvements stem from IID's ability to complete multiple editing instructions in a single pass through multi-instruction influence disentanglement, thereby circumventing degradation and artifacts caused by iterative denoising and repeated encoding-decoding through VAE, and mitigating the instruction conflict.

**Qualitative evaluation.** We present qualitative comparisons of our proposed IID method against Step-by-Step and NaiveCon baselines on our custom dataset, as shown in Fig. 4. The results demonstrate that IID consistently reduces visual distortion and degradation across all three DiT-based editing backbones. For instance, the sequential approach often introduces artifacts such as unrealistic color shifts, blurred backgrounds (e.g., in the top two rows), and significant noise, even using the advanced FluxKontext. In contrast, IID preserves high visual fidelity by processing

Table 2: Ablation study on of $S$, $A_{ZP}$ and $A_{ZZ}$ on MagicBrush dataset using Omnigen. $T = 50$.

| $A_{ZP}$ | $A_{ZZ}$ | $S$ | L1↓ | CLIP-I↑ | CLIP-T↑ |
|---|---|---|---|---|---|
| ✓ | | 45 | 0.1128 | 0.8599 | 0.2805 |
| | ✓ | 45 | 0.1132 | 0.8631 | 0.2818 |
| ✓ | | 49 | 0.1262 | 0.8439 | 0.2798 |
| ✓ | | 40 | 0.1087 | 0.8680 | 0.2824 |
| ✓ | | 30 | 0.1069 | 0.8715 | 0.2830 |
| ✓ | | 20 | 0.1061 | 0.8705 | 0.2866 |

all instructions in parallel in a single diffusion pass, thereby avoiding the cumulative errors inherent in iterative editing. Furthermore, IID achieves superior instruction completion where Step and NaiveCon frequently fail. This performance gain arises from IID's ability to generate accurate, instruction-aware masks that effectively disentangle the influence of each instruction, minimizing interference. By comparison, Step often disrupts spatial structure, while NaiveCon suffers from instruction conflicts during reverse diffusion, leading to incomplete edits. We also evaluate IID against NaiveCon using U-Net-based editing backbones in Fig 5. The results indicate that integrating IID with DiT-based models yields significantly better performance in terms of visual style, instruction completion, and content preservation, compared to U-Net-based models. These findings support the rationale and potential of our approach for multi-instruction editing on DiT-based architectures. For additional experiments on the MagicBrush dataset, please refer to Fig. 12 in the Appendix.

**Human Preference Study.** We conduct a human preference study on the MagicBrush and our custom dataset, as shown in Fig.6, and report the statistical significance in Fig.16. The results indicate that as the number of editing instructions increases, the human preference rate of IID steadily improves in both Instruction Alignment and Image Alignment. In contrast, the preference rates for the other two methods consistently decline. Specifically, the Step baseline shows a sharper drop in Image Alignment, while NaiveCon experiences a more significant decline in Instruction Alignment, particularly when the number of instructions exceeds three. These phenomena provide strong evidence of IID's superiority in terms of better instruction adherence and overall editing quality in multi-instruction guided image editing.

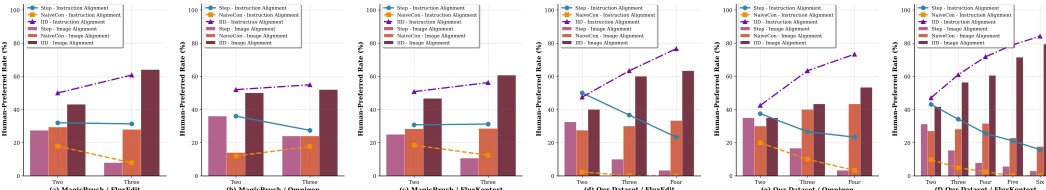

Figure 6: Human preference results on MagicBrush and our own dataset based on evaluations from 40 participants.. Human Preference rates reflect the proportion of times a method is selected as best under Instruction Alignment or Image Alignment, across different test settings defined by dataset and number of editing instructions.

## 5.3 ABLATION STUDY

First, we investigate the impact of the predefined step selection parameter $S$ and different mask extraction methods ($A_{ZP}$ and $A_{ZZ}$) on editing performance using the MagicBrush dataset in Table 2. The results reveal the following insights: 1) When comparing $A_{ZP}$ and $A_{ZZ}$ at step $S = 45$, both attention-based approaches exhibit comparable performance. Notably, $A_{ZZ}$ achieves slightly better CLIP-based scores, while $A_{ZP}$ yields a marginally lower L1 error. This indicates that both attention mechanisms are capable of effectively localizing the editing regions. 2) Regarding the influence of the

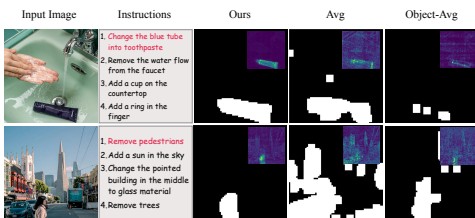

Figure 7: Ablation study of mask generation strategies on Omnigen at penultimate layer

step $S$, we observe a consistent trend that smaller step sizes tend to result in improved performance. Nonetheless, due to the parallel efficiency benefits associated with a larger step, we adopt $S = 45$ (0.9T) as the default configuration for mask extraction to strike a balance between performance and computational efficiency. Moreover, as shown in Table 5, IID demonstrates significant time efficiency as the number of instructions increases, achieving up to a 4× speedup on Omnigen and a 2× speedup on FluxKontext compared to sequential editing.

Second, we compare our proposed head-wise mask generation strategy with the *Avg* method that averages the attention map across all heads and applies the same binarization technique as the IID and the *Object-Avg*, which computes the average of cross-attention maps between the text token corresponding to the editable object and image tokens across all attention heads employed in U-Net-based editing models (Guo & Lin, 2024) in Fig. 7. Compared to *Avg*, our method

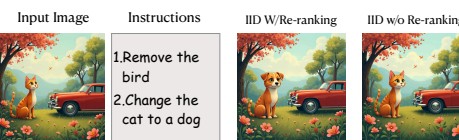

Figure 8: Ablation study of instruction re-ranking on FluxEdit.

effectively suppresses erroneous masks in non-editable regions, demonstrating the effectiveness of instruction-wise subtraction in filtering irrelevant noise. In contrast to *Object-Avg*, our results show that traditional U-Net-based mask generation is suboptimal for DiT-based models, likely due to the richer contextual information captured by self-attention between text and image tokens instead of the cross-attention alone.

Third, we examine the effectiveness of the influence score-based instruction re-ranking method proposed in the **Adaptive Blender** module of IID in Fig. 8 on FluxEdit. The results show that removing the re-ranking mechanism leads to failure in handling weaker instructions such as "Change the cat to a dog", highlighting the necessity of balancing the editing influence of instructions during parallel processing. Finally, additional ablation studies are provided in Appendix A.2.

## 6 CONCLUSION

In this work, we conduct an in-depth investigation of self-attention mechanisms in DiTs for instruction-guided image editing, uncovering key insights that could inform future research. Based on these findings, we introduce IID, a novel framework that enables parallel processing of multiple edits in

a single denoising process, reducing diffusion steps while improving editing quality compared to step-by-step editing. Our approach first localizes editing regions using a head-wise mask generation strategy at a predefined step. To achieve simultaneous editing, we adaptively concatenate instructions based on their influence scores and blend latent representations of multiple instructions to construct the noisy image input for the composite instruction. Finally, we construct an attention mask to mitigate instruction conflicts and let the editing model take these as input to denoise over subsequent timesteps. Extensive evaluations on an open-source multi-turn editing dataset and custom benchmarks demonstrate the effectiveness of our method.

## ETHICS STATEMENT

This work presents a novel framework that enables the parallel processing of multiple edits in a single denoising process, which significantly reduces diffusion steps and improves editing performance. While the technology democratizes advanced image manipulation and reduces computational overhead, it also raises ethical concerns regarding potential misuse, such as generating deceptive content or deepfakes. To mitigate such risks, we advocate for safeguards including provenance tracking, user authentication, and transparent disclosure of edits, along with policies that discourage malicious use. Our research is grounded in responsible practices, emphasizing transparency about limitations and promoting continued development of detection tools. We believe that, with appropriate safeguards, the societal benefits of accessible and efficient image editing can outweigh the risks, contributing meaningfully to ethical discourse surrounding AI-generated content.

## REPRODUCIBILITY STATEMENT

To ensure the reproducibility of our work, we provide comprehensive details in Appendix A.1 regarding datasets, baseline methods, human preference studies, and the implementation of our proposed IID framework across three DiT-based editing models (FluxEdit, Omnigen, and FluxKontext). Our approach demonstrates sufficient robustness, with the hyperparameter $S = 0.9T$ proving to be generalizable across different editing architectures. Finally, we will release our code upon acceptance of this paper.

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

# A APPENDIX

## A.1 MORE DETAILS ON EXPERIMENT SETTING

**Datasets** We verify the effectiveness of our proposed Instruction Influence Disentanglement (IID) on two multi-instruction guided image editing datasets: MagicBrush (Zhang et al., 2023) and our own constructed dataset. Statistics for both datasets are provided in Table 3. Each test example in MagicBrush includes the original image, a sequence of editing instructions, the ground truth image corresponding to each instruction, and captions for both the original and edited images. In contrast, examples in our own dataset contain only the original image and the editing instructions. For Kontext-Bench (Batifol et al., 2025), where a single image may be associated with more than 20 instructions, we randomly sample 20 editing sessions for each instruction count, ranging from two to six.

To construct our own dataset, we first randomly selected 5,00 images and their associated single editing instructions from two open-source datasets involving EmuEdit (Sheynin et al., 2024) and OmniEdit (Wei et al., 2024)) as well as from a proprietary dataset for real-world scenarios (excluding any data that cannot be publicly released). These test samples can cover a diverse range of editing tasks, such as object addition, removal, and style transformation, making our benchmark both diverse and representative of real-world demands.

Given a single image, we aim to generate $N$ instructions for it. Starting from the original instruction, we use GPT-4o[1] to generate additional complementary editing instructions. Specifically, GPT-4o is first prompted to detect editable objects in the image, then select $N$ objects that can be edited, and finally generate the unrepeated editing instruction for the selected object. Additionally, the model is asked to provide a rationale for each instruction, which helps filter out low-quality or irrelevant instructions. The exact prompting strategy is illustrated in the Fig. 10. Considering that some instructions significantly exceed the capabilities of Ominigen and FluxEdit, or are of low quality, we perform a two-stage evaluation for each test case. First, all editing instructions are processed by the editing model. Then, human experts manually assess the quality of the resulting outputs. If the overall error rate exceeds 60%, the case is considered beyond the model's ability and is excluded from further use. From the remaining qualified data, we randomly sample to construct the final dataset, maintaining an instruction count distribution ratio of 2:2:2:1:1. Illustrative examples of images and corresponding instructions from the three datasets are presented in Fig. 9.

Table 3: Statistics of multi-instruction editing datasets, including MagicBrush (excluding sessions with one Edit ) and Ours.

| Number of | MagicBrush | Ours | Kontext-Bench |
|---|---|---|---|
| - Sessions with Two Edits | 120 | 40 | 20 |
| - Sessions with Three Edits | 199 | 40 | 20 |
| - Sessions with Four Edits | 0 | 40 | 20 |
| - Sessions with Five Edits | 0 | 20 | 20 |
| - Sessions with Six Edits | 0 | 20 | 20 |
| Edit Turns | 319 | 160 | 100 |

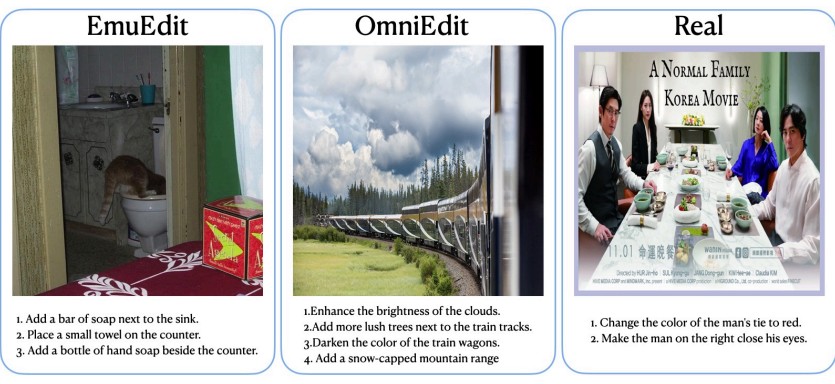

Figure 9: Examples of our constructed datasets.

---

[1] https://platform.openai.com/docs/models/gpt-4o

```
You are an assistant that only speaks JSON. Do not write normal text.
Your task is to generate {num_rounds} distinct image editing instructions, starting from an initial high-level instruction: "{instruction}".

Output a single JSON object with the exact following keys and structure:

- `"instruction_list"`: (List of Strings)
   - Content: Exactly {num_rounds} distinct editing instructions.
   - Requirements for each instruction:
      1. Must be a simple, clear sentence.
      2. Targets the corresponding object in the `"edited object"` string (see below, maintaining order).
      3. The instruction cannot be the same as the input

- `"detection result list"`: (String)
   - Content: A descriptive string that acts as a concise caption of the conceptual image. This string must also contain, typically as a comma-
separated list, at least {num_rounds} specific and distinct objects detected or inferred in the scene. These objects are candidates for editing. (This
field combines the idea of a caption/ORI PROMPT and a list of detected objects).

- `"edited object"`: (String)
   - Content: A single string containing a comma-separated list of exactly {num_rounds} objects.
   - Requirements for these objects:
      1. Each object must be chosen from the objects identified in your `"detection result list"` string.
      2. These are the objects targeted by the corresponding instructions in `"instruction_list"` (order matters).

- `"reason"`: (String)
   - Content: A single string containing {num_rounds} reasons, formatted as a numbered list (e.g., "1. Reason A. 2. Reason B. ...").
   - Each numbered reason corresponds to an instruction in `"instruction_list"` (and its targeted object) at the same position.
   - Each reason should briefly explain the rationale for choosing that specific object to edit and the nature of the editing instruction.

The initial context for your generation is the instruction: "{instruction}".
Ensure all {num_rounds} generated editing operations (object, instruction, reason) are unique.

Example for {num_rounds} = 2 (ensure your output is a single JSON string without comments):
{{
  "instruction_list": ["Make the cat black.", "Place a red ball near the cat."],
  "detection result list": "A scene with a white cat playing on a green rug, with a window in the background, and a bookshelf.",
  "edited object": "white cat, green rug",
  "reason": "1. Changing the cat's color is a simple modification. 2. Adding a ball provides an interactive element for the cat, using the rug as a
reference area."
}}
```

Figure 10: System prompt for generating multi-step editing instructions using GPT-4o.

**Baselines**   Given the lack of existing multi-instruction-guided image editing approaches specifically designed for DiT-based architectures, we evaluate our proposed framework, **IID**, against two baseline strategies: (1) step-by-step editing and (2) naive instruction concatenation. In the sequential editing setting, a sequence of instructions $\{P_1, \ldots, P_N\}$ is applied iteratively, where the noised image (prior to decoding by the VAE) from $P_{i-1}$ serves as input for the next instruction $P_i$. As such, this approach requires $N$ independent diffusion processes, one for each instruction. In contrast, the naive concatenation baseline merges all instructions into a single prompt by commas, guiding the editing model to perform the entire sequence in a single diffusion process.

For DiT-based instruction-guided image editing models, we consider FluxEdit (Paul, 2024), Omni-gen (Xiao et al., 2024) and FluxKontext (Batifol et al., 2025) which are representative implementations of the three DiT architectures proposed in (Peebles & Xie, 2023) (adaLN-Zero and In-Context Conditioning), respectively. In addition, we include target caption-guided editing models such as SD-SDEdit (Shi et al., 2024) and Null Text Inversion (NTI) (Mokady et al., 2023) as well as the multi-aspect parallel editing method ParaEdit (Huang et al., 2025). We also evaluate a range of U-Net-based instruction-guided editing models, including HIVE (Zhang et al., 2024), InstructPix2Pix (InP2P) (Brooks et al., 2023), HQ-EDIT (Hui et al., 2025), InP2P-Magic (Zhang et al., 2023) and SmartEdit (Huang et al., 2024). Except for ParaEdit, all of the aforementioned models are run in a step-by-step manner. Below, we provide concise descriptions of each of these baseline models. Beyond this work, a potential baseline could be FOI. However, due to the unavailability of its source code, we are unable to include it in our comparative analysis.

**SD-SDEdit** is a diffusion-based image synthesis and editing method that leverages a stochastic differential equation (SDE) prior to iteratively denoise a noisy version of the user-provided input image, thereby enhancing realism while preserving faithfulness to the guidance (e.g., hand-drawn strokes). Unlike GAN-based approaches, SD-SDEdit requires no task-specific training or inversion,

enabling general-purpose editing across diverse tasks such as stroke-based image synthesis, image editing, and compositing.

**NTI** is a text-based image editing method that enables high-fidelity manipulation of real images within pretrained diffusion models by introducing an accurate inversion technique. It features two key components: (i) pivotal inversion, which optimizes around a single pivotal noise vector per denoising step to anchor the inversion process; and (ii) NULL-text optimization, which modifies only the unconditional textual embedding used in classifier-free guidance, preserving both the model weights and conditional prompt embedding. This approach allows intuitive prompt-based editing without requiring model fine-tuning.

**ParaEdit** is a multi-attribute image caption-driven editing framework that enables simultaneous modification of multiple objects or attributes within a single editing pass. It introduces a multi-branch design with an attention distribution mechanism that allows parallel processing of edits across different semantic regions, preserving the quality of single edits while significantly enhancing performance in multi-instruction scenarios.

**HIVE**, is a framework for instructional image editing that leverages human feedback to learn user preferences through a reward function and fine-tunes U-Net-based diffusion models for improved alignment with editing instructions.

**InP2P**, is a conditional diffusion model trained on a large dataset generated by combining a language model (GPT-3) and a text-to-image model (Stable Diffusion). It edits images from human-written instructions directly in the forward pass without requiring per-example fine-tuning or inversion, enabling fast and generalizable image editing from instructions

**HQ-EDIT**, collects large-scale, high-quality instruction-based image editing data through a scalable pipeline that leverages foundation models such as GPT-4V and DALL·E 3 to filter low-quality samples and then finetune the editing models like InstructPix2Pix on this dataset, outperforming counterparts trained on human-annotated data.

**InP2P-Magic**, introduces MAGICBRUSH, the first large-scale, manually annotated dataset for instruction-guided real image editing, covering diverse scenarios such as single-turn, multi-turn, mask-provided, and mask-free editing, and then fine-tunes InstructPix2Pix on this dataset to improve editing quality.

**SmartEdit**, introduces MAGICBRUSH, the first large-scale, manually annotated dataset for instruction-guided real image editing, covering diverse scenarios such as single-turn, multi-turn, mask-provided, and mask-free editing,g and then fine-tune InstructPix2Pix on this dataset to improve editing quality.

**SmartEdit**,introduces a novel framework for complex instruction-based image editing by leveraging Multimodal Large Language Models (MLLMs) to enhance semantic understanding and visual reasoning. It proposes a Bidirectional Interaction Module to enable rich information exchange between the image and language modalities, and adopts a two-stage training strategy that incorporates perception data and a small amount of complex editing examples.

**Details of human preference study**   We conducted a comprehensive human evaluation to assess the quality of multi-instruction image editing. Forty participants with backgrounds in computer vision and image editing were recruited to evaluate editing results. Each participant was presented with the original image, a sequence of editing instructions, and the edited results from different methods. For each dataset, we randomly sampled 100 editing sessions for each participant.

The evaluation was conducted along two primary dimensions: **instruction alignment** and **image alignment**. For instruction alignment, participants assessed the quality of edits based on two criteria: semantic accuracy, which measures whether each edit accurately reflects the intended modifications specified in the instructions, and completion rate, defined as the proportion of instructions that were successfully handling. A higher completion rate indicates stronger adherence to the provided instructions. Participants were explicitly instructed to prioritize results that implemented a greater number of editing instructions when evaluating overall quality. For image alignment, the focus was on the preservation of the original image characteristics after editing. This was assessed through three aspects: structural preservation, referring to the maintenance of key structural elements from the original image; texture fidelity, evaluating whether the original textures and patterns were retained

in unedited regions; and color consistency, which measures the extent to which the original color scheme was preserved in areas not targeted by the editing instructions.

To ensure reliable and consistent evaluation, participants followed a set of strict assessment guidelines. They were required to carefully review all editing instructions prior to scoring, consider both the accuracy of local edits and the overall visual quality of the image, and make a single-choice selection to identify the best-performing method under each evaluation dimension. The final human preference score for each method was computed as the proportion of times it was selected as the best method, divided by the total number of evaluation instances in one specific setting. For example, under the two-instruction setting for Instruction Alignment, if the number of wins for OmniGen w/ Step, OmniGen w/ NaiveCon, and OmniGen w/ IID were 10, 10, and 30 respectively, their corresponding human preference scores would be 0.2, 0.2, and 0.6. We present our evaluation interface in Fig. 11.

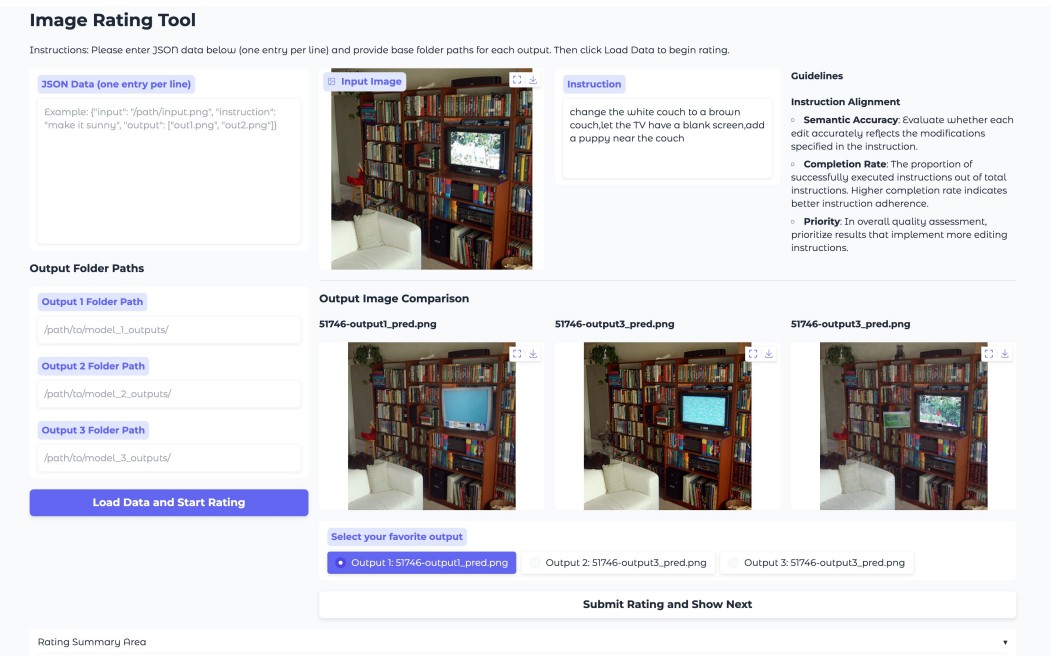

Figure 11: Demonstration of our system for human preference study.

**Implementation details**   We utilize three DiT-based instruction-guided image editing models[2] including FluxEdit, Omnigen and FLuxKontext to verify the effectiveness of our proposed IID framework. Since the original FluxEdit exhibits suboptimal performance on common editing tasks such as object addition, deletion, and modification, we fine-tune FLUX.1-dev[3] using the same control framework as FluxEdit on a private dataset comprising 300K high-quality instruction–image pairs. We plan to release the fine-tuned checkpoint in the near future. For OmniGen, we adopt the implementation available at `https://huggingface.co/spaces/Shitao/OmniGen`. We note that this version differs from the GitHub implementation[4] in how the query matrix $Q$ is constructed within the self-attention mechanism of DiT. Specifically, the Hugging Face (diffusers) version concatenates the instruction tokens, source image tokens, timestep token, and noisy image tokens to form $Q$ across all diffusion steps. In contrast, the GitHub version performs concatenation of all conditioning tokens and noisy image tokens only at the initial timestep ($t = T$), and constructs $Q$ by concatenating the timestep token with noisy image tokens. We argue that the diffuser-based implementation reflects a more principled design.

In all experiments, we fix the hyperparameters related to image generation for both models. Concretely, for FluxEdit, we set the text guidance scale to 30, the maximum instruction token length to

---

[2]`https://huggingface.co/black-forest-labs/FLUX.1-Kontext-dev`

[3]`https://huggingface.co/black-forest-labs/FLUX.1-dev`

[4]`https://github.com/VectorSpaceLab/OmniGen`

Table 4: All instructions are executed in a step by step manner on the MagicBrush dataset. All evaluation metrics are computed as the difference between the metric value of the final instruction's editing result and the average metric value across all editing rounds.

| Method | L1↓ | L2↓ | CLIP-I↑ | DINO↑ | CLIP-T↑ |
|---|---|---|---|---|---|
| *Instruction-guided* | | | | | |
| HIVE (Zhang et al., 2024) | 0.0393 | 0.0212 | -0.0512 | -0.1074 | -0.0092 |
| InP2P (Brooks et al., 2023) | 0.0372 | 0.0218 | -0.0266 | -0.0669 | -0.0061 |
| HQ-EDIT (Hui et al., 2025) | 0.0229 | 0.0152 | -0.0218 | -0.0510 | -0.0014 |
| InP2P-Magic (Zhang et al., 2023) | 0.0303 | 0.0146 | -0.0334 | -0.0621 | -0.0029 |
| Omnigen (Xiao et al., 2024) | 0.0311 | 0.0168 | -0.0326 | -0.0600 | -0.0005 |
| FluxEdit | 0.0291 | 0.0115 | -0.0419 | -0.0786 | -0.0051 |

512, and the total diffusion steps to 30 ($T = 30$). For OmniGen, we set the text guidance scale to 2.5, the image guidance scale to 1.6, and the total diffusion steps to 50 ($T = 50$). For FLuxKontext, we set the guidance scale to 2.5, the total diffusion steps to 28 ($T = 28$) and $S = 25$. To maintain consistency and ensure the general applicability of our IID framework across models with different diffusion steps, we set the predefined step as $S = 0.9 \times T$ without requiring extensive hyperparameter tuning. This validates the robustness and generality of IID. For both models, the editing mask is extracted from the attention map $A_{ZP}$ in the penultimate transformer layer. In addition, for all other baseline models, we follow their publicly available implementations. While in typical multi-instruction parallel editing scenarios, each instruction is expected to focus on a distinct editing target. However, due to the suboptimal quality of some generated instructions, overlapping masks may still occur. To address such mask conflicts, we also apply Algorithm 1 with a conflict threshold $\theta = 0$, ensuring that only non-overlapping instruction groups are processed in each iteration.

## A.2 More Qualitative and Quantitative evaluation

### A.2.1 Quantitative evaluation

For multi-instruction guided image editing tasks, we aim to quantitatively validate the hypothesis that step-by-step editing may lead to a degradation in image quality, such as increased artifacts or noise accumulation. As shown in Table 4, we compute the difference between the metric value of the final instruction's editing result and the average metric value across all editing rounds. The results demonstrate that, across all evaluation metrics, the average performance over all rounds consistently outperforms that of the final round. This provides empirical evidence supporting the argument that sequential processing of instructions can lead to cumulative quality degradation in the edited images.

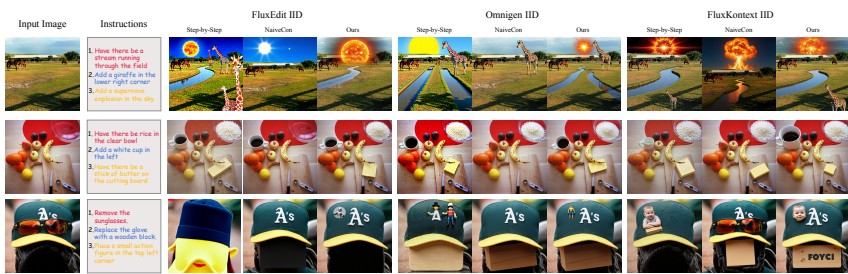

Figure 12: Qualitative comparison results of our proposed IID against Step-by-Step and NaiveCon on FluxEdit, Omnigen, and FluxKontext, respectively.

### A.2.2 Ablation of Extracted Layers

As illustrated in Fig. 14, we extract masks for each instruction from different layers of the editing models. The results indicate that for both editing models, lower layers are not suitable for mask extraction, as information tends to interact across the image, resulting in dispersed attention weights. As the layer depth increases, the semantic information of different tokens gradually converges to form higher-level semantic features, enabling a more focused attention on the edited regions. Notably, the masks extracted from the final layer exhibit lower quality. We suggest that this may be because

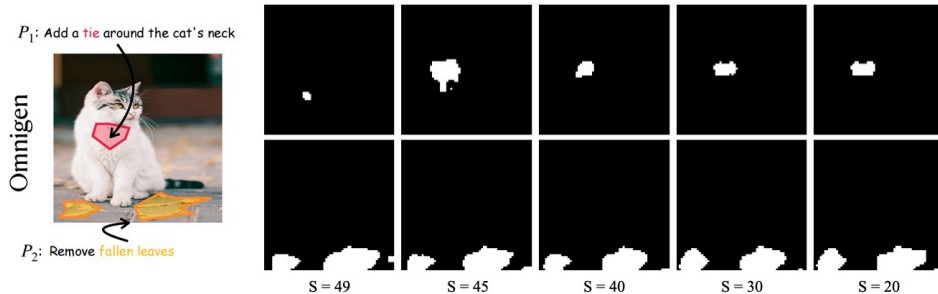

Figure 13: Ablation study on the influence of the step used for mask extraction on the quality of the generated masks.

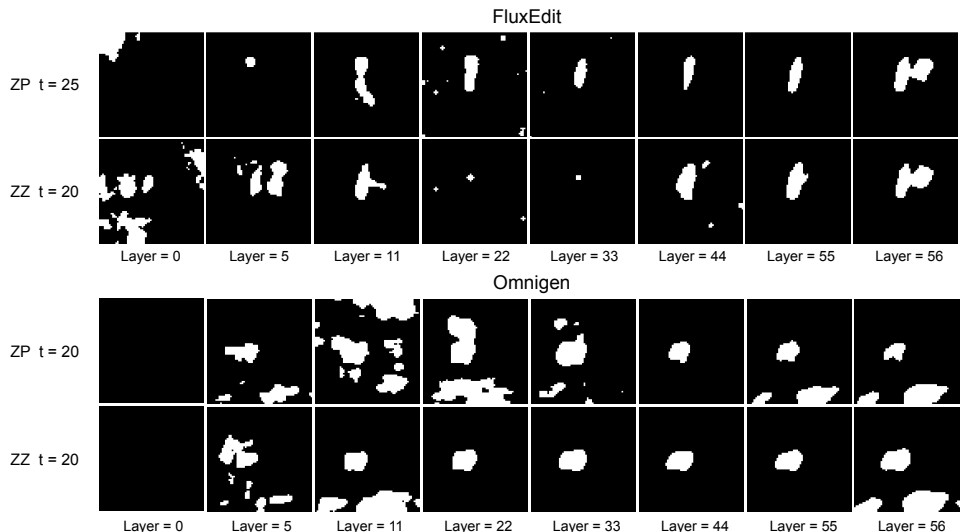

Figure 14: Ablation study on the influence of the layer used for mask extraction on the quality of the generated masks.

the feature of the final layer is utilized for image reconstruction through the VAE, making it less effective for mask generation. A similar phenomenon has been observed in transformer-based text classification models, where the final layer features are not always ideal for semantic representation (Liu et al., 2024), and the textual features also demonstrate a hierarchical pattern (Jawahar et al., 2019). Therefore, we use the penultimate layer to extract masks in our main experiments.

### A.2.3 ABLATION OF THE TIMESTEP ON MASK GENERATION

As demonstrated in Fig. 13, we extract masks from the penultimate layer of both editing models at different timesteps. Generally, as the diffusion steps progress, the quality of the masks improves. After a certain number of steps, the changes in the masks become minimal, and the results stabilize.

### A.2.4 ABLATION OF MASK COMPUTATION SETTINGS

One of our key contributions lies in the head-wise mask extraction strategy (Eq. 3) designed for DiT-based diffusion models, motivated by the attention-based interpretability analysis in Sec. 4.1. While our default mask refinement pipeline, comprising a Gaussian filter followed by Otsu's thresholding (Eq. 4) has proven effective, these components can be replaced with alternative engineering strategies.

In our approach, the Gaussian filter is applied to suppress high-frequency noise, ensuring a smoother and more coherent mask. Otsu's filter then adaptively determines an optimal threshold that maximizes inter-class variance, allowing for effective separation of foreground and background without manual

Table 5: Time cost across different instruction counts ($N$) using FluxKontext and Omnigen.

| Dataset | Method | N=2 | N=3 | N=4 | N=5 | N=6 |
|---------|--------|-----|-----|-----|-----|-----|
| **FluxKontext (Time Cost in S)** | | | | | | |
| | Step-by-Step | 29.12 | 43.64 | 59.12 | 72.68 | 88.36 |
| | Ours | 22.67 | 33.10 | 40.41 | 44.23 | 47.67 |
| **Omnigen (Time Cost in S)** | | | | | | |
| | Step-by-Step | 134.56 | 201.73 | 268.86 | 335.12 | 408.12 |
| | Ours | 79.93 | 87.56 | 95.23 | 103.45 | 110.98 |

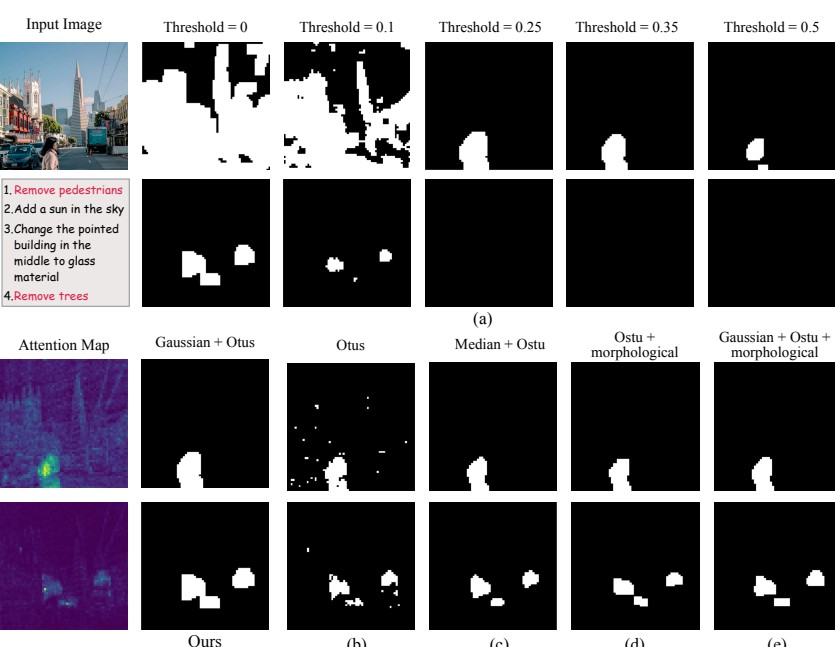

Figure 15: Illustrative examples of different mask computation settings.

tuning. To investigate of different mask computation settings, we qualitatively compare several alternative configurations in Fig. 15: (a) Gaussian filter followed by multiple fixed thresholds for binarization (b) Otsu's filter without Gaussian smoothing, (c) median filtering followed by Otsu's filter, (d) Otsu's filter with morphological operations (specifically, erosion followed by dilation), and (e) Gaussian filter combined with Otsu's filter and morphological operations.

Our findings are as follows. First, using fixed thresholds is highly sensitive to the specific test case, as the value ranges of attention maps vary significantly across instructions, even after Gaussian smoothing, thus rendering fixed thresholds unsuitable for general use. In contrast, Otsu's filter effectively addresses this issue by providing adaptive thresholding, thus improving generalization. Second, morphological operations and median filtering can serve a similar role to Gaussian filtering by suppressing high-frequency noise, resulting in cleaner masks with reduced artifacts. Lastly, we find that the combination of Gaussian filter and Otsu's thresholding already produces high-quality masks in most cases, often making additional morphological operations unnecessary.

### A.2.5 SUPPLEMENTARY EXPERIMENTS OF HUMAN STUDY

We conducted a comprehensive human evaluation to assess the quality of multi-instruction image editing by recruiting 40 participants to evaluate editing results across three datasets. The human preference rates for MagicBrush and our own dataset on FluxEdit, Omnigen, and FluxKontext are detailed in Fig.6 in the main body, while the results for Kontext-Bench (Batifol et al., 2025) on

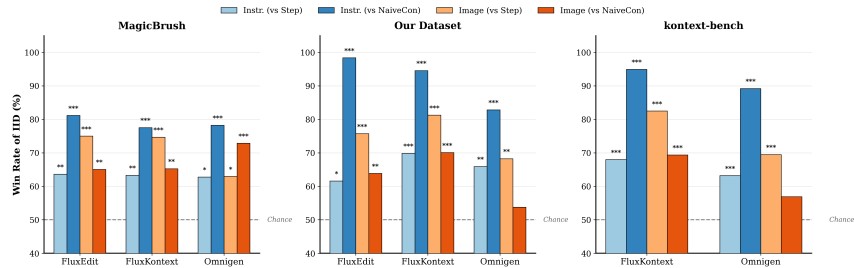

Figure 16: The win rate of our proposed IID against two baselines, Step and NaiveCon, along with statistical significance assessed using the Exact Binomial Test (*, **, *** denote statistically significant differences at p < 0.05, 0.01, and 0.001, respectively).

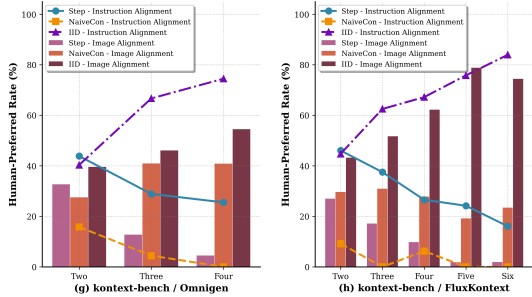

Figure 17: Human preference results on Kontext-Bench on Omnigen and FluxKontext.

Omnigen and FluxKontext are shown in Fig.17. Across different datasets and backbone editing models, the results consistently indicate that as the number of editing instructions increases, the human preference rate for IID steadily improves in both Instruction Alignment and Image Alignment. In contrast, preference rates for the other two methods, Step and NaiveCon, consistently decline. Additionally, we calculate statistical significance using the Exact Binomial Test, as presented in Fig. 16. The results show that IID significantly outperforms NaiveCon in Instruction Alignment, and as the number of instructions increases, it also significantly surpasses Step. Similarly, for Image Alignment, IID consistently outperforms Step, achieving win rates of approximately 70%–80%, and also performs better than NaiveCon.

### A.2.6   ANALYSIS OF FAILURE CASES

We conduct a failure case analysis of our proposed IID framework and identify two primary sources of failure. The first arises from conflicts in editing regions among multiple instructions. As shown in the second row of Fig. 18, the overlap between the headband and the laptop leads to the failure of the instruction to remove the headband. Similarly, in the third row, the flower and the wooden board share overlapping regions, resulting in the failure of the instruction to add a flower.

The second type of failure stems from individual instructions that exceed the model's inherent capabilities. For example, in the first row of Fig. 18, even when executed alone, the instruction to remove the milk box cannot be completed by OmniGen. Consequently, when multiple instructions are processed, the model still fails to carry out this specific operation.

The first failure type is arguably reasonable. As analyzed in Appendix A.4.2, such instruction conflicts may result from velocity misalignment between editing operations and the lack of multi-instruction training data. Two potential solutions exist: (1) our IID framework, which isolates instructions via masks during the diffusion process, and (2) synthesizing high-quality, complex instruction–image pairs, for which IID itself can serve as a data generator to improve the underlying editing model. However, in cases of overlapping target region among different instructions, the first strategy becomes ineffective because instructions can still attend to regions associated with other instructions. In such cases, the model's intrinsic capacity becomes the dominant factor, and averaging strategies offer limited benefit. For example, as shown in Fig. 19, the stronger model, FluxKontext, successfully

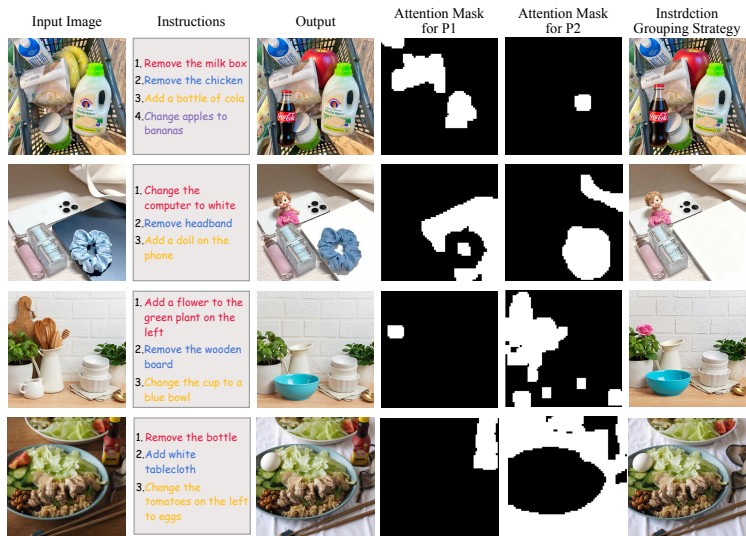

Figure 18: Failure case analysis of our proposed IID on Omnigen. The Instruction Grouping Strategy refers to using our automatic scheduling method for resolving mask conflicts in Appendix A.3.

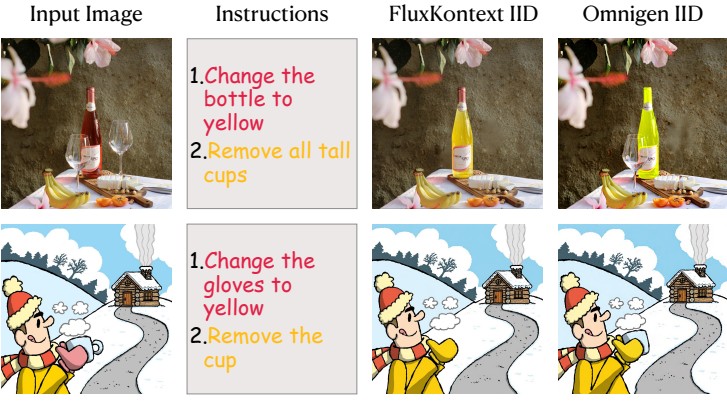

Figure 19: Failure case analysis of our proposed IID on Omnigen and FluxKontext.

completes both conflicting instructions, whereas the weaker OmniGen completes only one, providing empirical support for our hypothesis.

To further enhance IID's robustness in scenarios involving overlapping masks, we introduce an Optimal Instruction Grouping strategy in Appendix A.3. This approach partitions instructions into non-conflicting groups and schedules them to minimize the number of editing passes. As shown in the last column of Fig. 18, this strategy effectively resolves most of the failure cases.

Table 6: Quantitative experiments using different mask extraction strategies on the MagicBrush dataset.

| Method | L1↓ | L2↓ | CLIP-I↑ | DINO↑ | CLIP-T↑ |
|---|---|---|---|---|---|
| *Instruction-guided* | | | | | |
| IID | 0.1128 | 0.0466 | 0.8599 | 0.7604 | 0.2805 |
| SAM | 0.1131 | 0.0448 | 0.8487 | 0.7541 | 0.2783 |
| *Avg* | 0.1132 | 0.0472 | 0.8386 | 0.7401 | 0.2742 |
| *Object-Avg* | 0.1130 | 0.0469 | 0.8393 | 0.7428 | 0.2763 |

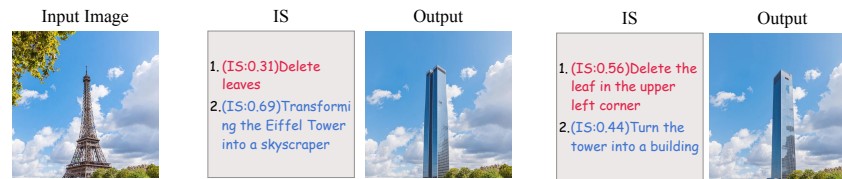

Figure 20: Illustrative examples of the impact of instruction phrasing on the scale of Influence Scores.

### A.2.7 ANALYSIS OF INFLUENCE SCORE (IS)

In the Adaptive-Blender component of IID, we propose computing the Influence Score (IS) for each instruction to quantify its relative impact on the editing process. The instructions are then sorted in ascending order based on their IS values to approximately equalize their influence during editing. To examine how the paraphrasing of instructions affects the scale of IS, we present a qualitative example in Fig. 20. The results show that rephrasing an instruction can lead to differences in its IS; however, this does not affect the final editing result. For example, in the right of Fig. 20, the IS of the first instruction increases compared to the middle one, but the re-ranking strategy ensures that all instructions are still executed appropriately.

### A.2.8 ABLATION OF MASK EXTRACTION STRATEGIES

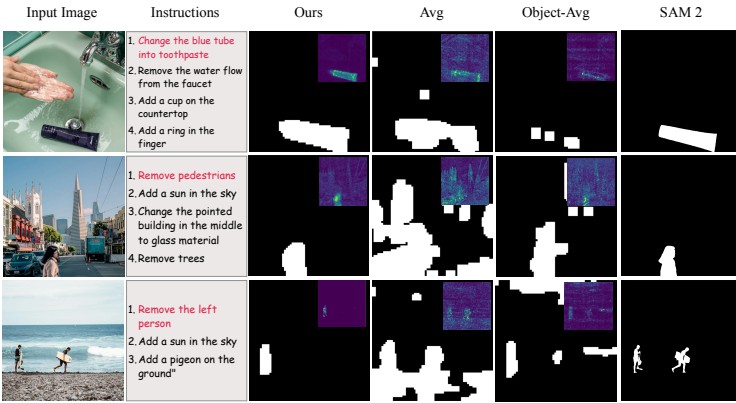

Figure 21: Ablation study of mask generation strategies on Omnigen.

**SAM.** In Sec. 5.3 and Fig. 7, we have compared our proposed head-wise mask generation strategy against *Avg* that averages the attention map across all heads and applies the same binarization technique as the IID and the *Object-Avg*, which computes the average of cross-attention maps between the text token corresponding to the editable object and image tokens across all attention heads employed in FOI (Guo & Lin, 2024). We further compare our mask extraction strategy with a SAM-based pipeline, which first employs Grounding DINO to identify the bounding box of the target object of the given editing instructions, followed by SAM for semantic segmentation to generate the corresponding mask. As shown in Fig. 21, for relatively simple editing tasks such as object removal, this approach is capable of producing high-quality masks. However, the SAM-based pipeline requires explicit specification of the object to be segmented and thus struggles with instructions involving non-existent elements, such as object addition. Moreover, it is limited in handling background modifications and cannot accurately interpret complex instructions like spatial constraints or specific quantities.

**Experiments on MagicBrush dataset.** We qualitatively compare different mask extraction strategies in Table 6. For SAM, if the target editing object is not present in the image (e.g., adding operation), a random mask is generated. The results show that our proposed head-wise mask extrac-

tion method not only better preserves the original image content but also more effectively fulfills the editing instructions.

### A.2.9 ANALYSIS OF FOI

As shown in Fig.5, the mask strategy used by FOI (Guo & Lin, 2024) fails to accurately localize the target editing regions specified by different instructions. We adapt FOT from U-Net-based diffusion models to DiT and report its performance on the MagicBrush dataset (Omnigen) in Table 1. The results demonstrate that IID significantly outperforms FOI. A qualitative comparison between FOI and IID is provided in Fig. 5, where FOI often fails to complete all the given instructions and produces unrealistic objects, such as windows in the final column of Fig.5.

### A.2.10 ABLATION OF EACH COMPONENT IN IID

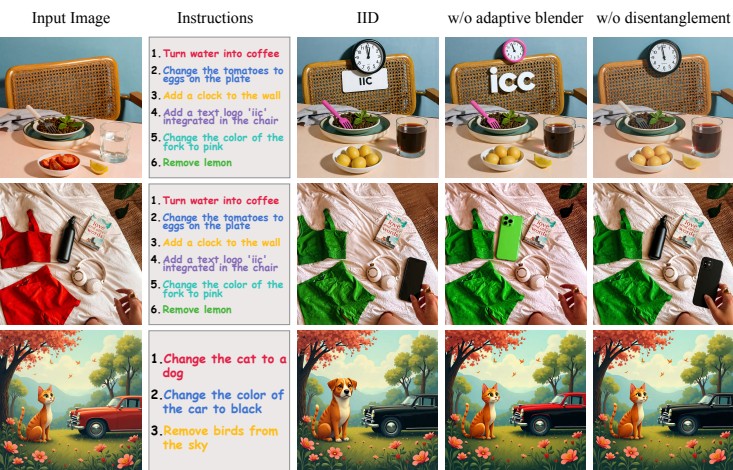

Figure 22: Ablation study of each component of IID on Omnigen.

For multi-instruction image editing, sequential editing often leads to cumulative distortions and quality degradation, while naive concatenation of instructions frequently fails to fulfill all intended modifications due to instruction conflicts. To address these challenges, we propose the IID framework, which aims to disentangle the influence of multiple instructions and complete all instructions in a single editing turn.

IID consists of three essential components, involving Head-wise Mask Generation, Adaptive Blender, and Multi-instruction Influence Disentanglement. The first component generates accurate masks for each instruction. The second combines the text tokens and noisy image tokens of all instructions into a compositional representation, enabling parallel execution of instructions in a single editing turn. The third component disentangles the influence of each instruction, thus mitigating instruction conflicts and completing all intended notifications. The output of each component feeds into the next, and all are necessary to achieve effective multi-instruction editing.

While various mask generation strategies and settings have been investigated in Section 5.3, Appendix A.2.4, and Appendix A.2.8, we focus on evaluating the contributions of Adaptive Blender and Multi-instruction Influence Disentanglement in Fig. 22. Since Adaptive Blender is critical for enabling parallel editing, we assess its role by replacing the mask-based blending strategy in Eq (5) with simple averaging, denoted as w/o Adaptive Blender, while keeping the remaining components unchanged. Similarly, we evaluate the effect of removing the Influence Disentanglement component by allowing each instruction to attend to the entire image. As illustrated in Fig. 22, replacing Adaptive Blender with averaging leads to ambiguous edits, where visual attributes intended for one region spill into others. For example, the clock inherits the color meant for the fork in first row of Fig. 22,. Without the Influence Disentanglement module, instruction conflicts emerge, preventing the model from fulfilling all instructed edits.

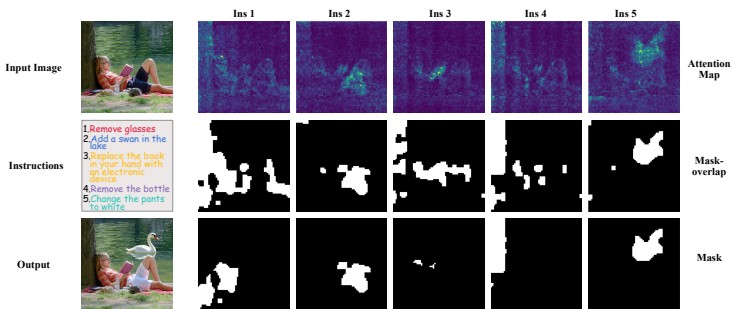

Figure 23: Analysis of removing overlapping regions in the *Avg* Mask Extraction Strategy. Each original mask for each instruction has center point, and overlapping pixels are assigned to the mask whose center is nearest.

### A.3 AUTOMATIC SCHEDULE STRATEGY ON MASK CONFLICTS

In typical multi-instruction parallel editing scenarios, each instruction is expected to focus on a distinct editing target, rather than applying multiple operation to a single object. To enhance the generality and robustness of the IID framework, we wish it to also accommodate such challenging cases. Additionally, when the target objects are spatially adjacent or when one of the instructions involves a global operation, the corresponding instruction masks may exhibit significant overlap. While IID possesses a built-in mechanism to alleviate mask conflicts by averaging the latent representations $z_{S,i}$ across overlapping regions to incorporate the influence of all relevant instructions, the severity of instruction conflicts increases with the extent of the overlap due to the compounded effects of multiple instructions, leading to low-quality editing results.

To mitigate this issue, we introduce an **automatic scheduling strategy** that partitions the instruction set into multiple groups of non-conflicting instructions. In each editing iteration, only one group is processed to avoid interference, and the output of the current group serves as the input for the subsequent instruction group. When conflicts exist among a set of $N$ instructions, the scheduling problem can be formalized as a variant of the graph coloring problem, where each instruction is modeled as a node, and conflicts are represented as edges. The goal is to assign the smallest number of colors such that adjacent nodes (conflicting instructions) receive different colors, with each color denoting an independent execution batch. We employ a backtracking algorithm to obtain an **optimal coloring solution**, thereby minimizing the total number of required editing passes. The whole algorithm is detailed in Algorithm 1. We set the threshold as 0 ($\theta = 0$).

---

**Algorithm 1** Optimal Instruction Grouping for Mask-Conflict-Free Schedule

---

**Require:**
1: $N$ instructions: $\{P_1, P_2, ..., P_N\}$
2: Masks of instructions: $\{M_1, M_2, ..., M_N\}$
3: IoU threshold $\theta$
**Ensure:** Optimal grouping schedule $S$
4: **function** COMPUTECONFLICT($M_1, M_2$)
5:     $intersection \leftarrow |M_1 \cap M_2|$
6:     $union \leftarrow |M_1 \cup M_2|$
7:     **if** $union = 0$ **then return** 0
8:     **end if**
9:     $IoU \leftarrow intersection/union$ **return** $IoU > \theta$
10: **end function**
11: **function** BUILDCONFLICTMATRIX($\{M_1, ..., M_N\}$)
12:     $C \leftarrow N \times N$ matrix initialized with 0
13:     **for** $i \leftarrow 1$ to $N$ **do**
14:         **for** $j \leftarrow i + 1$ to $N$ **do**
15:             **if** COMPUTECONFLICT($M_i, M_j$) **then**
16:                 $C[i][j] \leftarrow 1$
17:                 $C[j][i] \leftarrow 1$
18:             **end if**
19:         **end for**
20:     **end forreturn** $C$
21: **end function**
22: **function** ISSAFE($node, color, colors, C$)
23:     **for** $neighbor \leftarrow 1$ to $N$ **do**
24:         **if** $C[node][neighbor] = 1$ **and** $colors[neighbor] = color$ **then return** false
25:         **end if**
26:     **end forreturn** true
27: **end function**
28: **function** BACKTRACK($node, colors, max\_color, C$)
29:     **if** $node = N$ **then return** true
30:     **end if**
31:     **for** $color \leftarrow 0$ to $max\_color - 1$ **do**
32:         **if** ISSAFE($node, color, colors, C$) **then**
33:             $colors[node] \leftarrow color$
34:             **if** BACKTRACK($node + 1, colors, max\_color, C$) **then return** true
35:             **end if**
36:             $colors[node] \leftarrow -1$         ▷ Backtrack
37:         **end if**
38:     **end forreturn** false
39: **end function**
40: **function** MINCOLORING($C$)
41:     **for** $max\_color \leftarrow 1$ to $N$ **do**
42:         $colors \leftarrow [-1, ..., -1]$         ▷ Length N
43:         **if** BACKTRACK($0, colors, max\_color, C$) **then**
44:             $S \leftarrow$ Group instructions by colors **return** $S$
45:         **end if**
46:     **end for**
47: **end function**
48: $C \leftarrow$ BUILDCONFLICTMATRIX($\{M_1, ..., M_N\}$)
49: $S \leftarrow$ MINCOLORING($C$) **return** $S$

---

## A.4 THEORETICAL ANALYSIS

### A.4.1 ERROR ACCUMULATION IN MULTI-ROUND RECTIFIED-FLOW EDITING

We give a concise proof that the numerical (and conditional) error in rectified-flow editing inevitably increases with the number of editing rounds. Assume we have $N$ instructions. The notation is identical to Sec. 3.

**Basic Assumptions**

1. **Lipschitz vector field.** For any $t \in [0, 1]$
$$\left\| \mathbf{v}_\theta(\mathbf{z}_1, t) - \mathbf{v}_\theta(\mathbf{z}_2, t) \right\| \leq L \left\| \mathbf{z}_1 - \mathbf{z}_2 \right\|.$$

2. **One–round duration.** Each editing round diffuses from $t = 0$ to $t = \tau$ and back, so the total physical time is $T := 2\tau$.

3. $s$-**th–order explicit solver.** Using step size $\Delta t$ (e.g. RK4 $\Rightarrow s = 4$), the classical global error bound for ODEs (cf. Iserles (2009)) gives

$$\boxed{\|e(T)\| \ \leq \ e^{LT} \|e(0)\| + C\, T\, \Delta t^s} \tag{6}$$

where $e(t)$ is the state error and $C$ depends on higher derivatives of $\mathbf{v}_\theta$.

Intuitively, $e^{LT} > 1$ amplifies the incoming error, while $C\, T\, \Delta t^s$ is the fresh error produced by this segment.

**From One Round to Many.** Define two positive constants
$$A := e^{LT} \ > 1, \quad B := C\, T\, \Delta t^s.$$

Applying Eq. (6) to one forward–backward pass yields
$$E_1 \ \leq \ A\, E_0 + B, \tag{7}$$
where $E_n$ is the error at the end of round $n$.

If the $(n-1)$-th round ends with error $E_{n-1}$, the next round produces
$$E_n \ \leq \ A\, E_{n-1} + B. \tag{8}$$

Iterating the linear recurrence Eq. (8) for $N$ rounds (or summing a length-$N$ non-homogeneous geometric sequence) gives

$$\boxed{E_N \ \leq \ A^N E_0 \ + \ B\, \frac{A^N - 1}{A - 1}} \tag{9}$$

**Implications.**

1. Because $A = e^{LT} > 1$, both terms on the right of Eq. 9 grow exponentially with $N$. Even if the initial error $E_0 = 0$, repeated numerical integration enlarges the newly introduced error $B$.

2. Any additional per-round guidance bias $\varepsilon_{\text{cond}}^{(N)}$ can be absorbed into $B$ and therefore accumulates likewise.

Hence the total error $E_N$ is a monotonically increasing function of the number of editing rounds, explaining the observed quality degradation in multi-step rectified-flow editing pipelines.

**Effectiveness of IID.** Based on the preceding error analysis, IID mitigates the exponential error accumulation by consolidating multiple editing instructions into a single composite instruction at the $S$-th step. This approach fundamentally circumvents the growth of accumulated error $E$ with respect to the number of rounds $K$, thereby substantially reducing noise artifacts and preserving fidelity in the edited images. More formally, instead of applying $N$ sequential editing rounds where error grows as $E_N \leq A^N E_0 + B\frac{A^N - 1}{A-1}$, IID performs instruction fusion to create a unified editing objective. This consolidated approach reduces the effective number of integration rounds from $N$ to 1, effectively bounding the error growth and maintaining superior image quality throughout the editing process.

### A.4.2 INSTRUCTION CONFLICT IN NAIVE CONCATENATION FOR MULTI-INSTRUCTION GUIDED IMAGE EDITING

Consider concatenating $N$ textual instructions $\mathbf{P} = [P_1, \ldots, P_N]$ into one prompt and sampling with the rectified flow ODE

$$\frac{dz}{dt} = \mathbf{v}_\theta\big(z_t, t, c_I, c_{\mathbf{P}}\big), \tag{10}$$

where $\mathbf{v}_\theta$ is the conditional velocity field, $c_I$ is the encoded source image, and $c_{\mathbf{P}}$ is the joint text embedding of all instructions. In practice, only 1–2 instructions are faithfully completed. Below, we summarize two complementary reasons.

**Dynamical Conflicts Are Exponentially Amplified.** Write the ideal velocity that would satisfy every $P_i$ by $\mathbf{v}^\star(z_t, t)$ and decompose the model field as

$$\mathbf{v}_\theta = \mathbf{v}^\star + \boldsymbol{\varepsilon}, \quad \|\boldsymbol{\varepsilon}(z_t, t)\| = \text{alignment error}.$$

With the same Lipschitz constant $L$ as in Sec. A.4.1, the variation-of-constants formula yields

$$\|\mathbf{e}(1)\| \ \leq \ \int_0^1 \big\|\boldsymbol{\varepsilon}(t)\big\| \exp\!\Big(\int_t^1 L(\tau)\,d\tau\Big)\,dt. \tag{11}$$

Therefore, we suggest the effect of many instructions as follows:

- When $N$ grows, the desired directions $P_i$ may become mutually inconsistent and the constant $\|\boldsymbol{\varepsilon}(t)\|$ then grows. For example, the instructions may have dual role in enforcing edits on target regions while preserving the unedited areas. When multiple instructions are processed simultaneously, they are mutually inconsistent and tend to compete for dominance, and stronger ones may dominate weaker ones, leading to incomplete editing.
- In rectified flow $L \gtrsim 1$, so the exponential kernel in Eq. 11 strongly amplifies even small misalignment. The solver therefore converges to a trajectory that compromises between conflicting $P_i$ and ends up only with a few dominant edits.

**Multi-instruction Data Become Sparse** Let $\mathcal{S}_i$ be the set of images satisfying $P_i$ and assume $\Pr(\mathbf{x} \in \mathcal{S}_i) = \alpha < 1$ and the events are approximately independent. Then

$$\Pr\big(\mathbf{x} \in \bigcap_{i=1}^N \mathcal{S}_i\big) = \alpha^N \ \xrightarrow[N \to \infty]{} \ 0, \tag{12}$$

i.e., training images that satisfy all instructions seldom appear when $N$ is large. Because recent work (Hertz et al., 2022) trains new models using synthetic data generated by editing models, since creating data through Photoshop is difficult to scale. However, these data-generating models themselves cannot complete multiple instructions, which may have made multi-instruction editing training data even more scarce.

**Effectiveness of IID.** Based on the above analysis, after concatenating multiple instructions into a composite one, IID uses masks to ensure that each instruction does not interfere with others during the diffusion processing, preventing interference with the editing regions of other instructions and reducing the growth of $|\varepsilon(t)|$. Moreover, IID provides a feasible method for generating multi-instruction data in the future. Additionally, training with high-quality and complex instructions also helps alleviate instruction conflicts. Although Batifol et al. (2025) did not release their data publicly, it can be anticipated that they adopted more complex instruction-image pairs as training data, based on insights from previous work (Xiao et al., 2024).

### A.5 MORE VISUALIZATION ON ATTENTION WEIGHT ANALYSIS IN DITS

We provide visualization of the attention maps $\bar{A}_{ZP_1}^j$ across all attention heads, corresponding to prompt $P_1$ for the cat image in Fig. 3. in Fig. 24, the visualization of the attention maps $\bar{A}_{ZP_2}^j$ across all attention heads, corresponding to prompt $P_2$ for the cat image in Fig. 3 in Fig. 25 and their difference in Fig. 26 and Fig. 27.

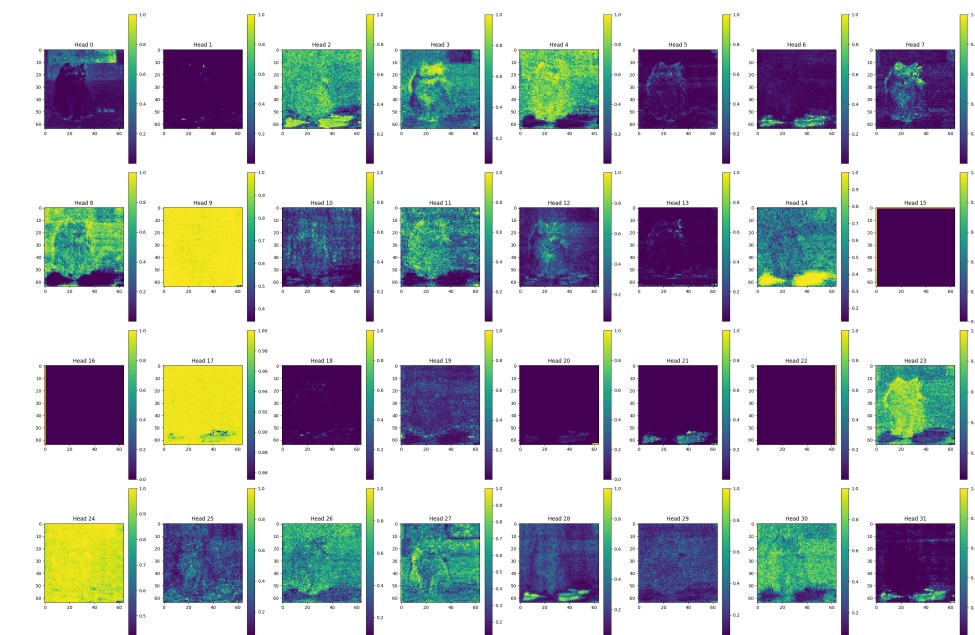

Figure 24: Visualization of the attention maps $\bar{A}_{ZP_1}^j$ across all attention heads, corresponding to prompt $P_1$ for the cat image in Fig. 3.

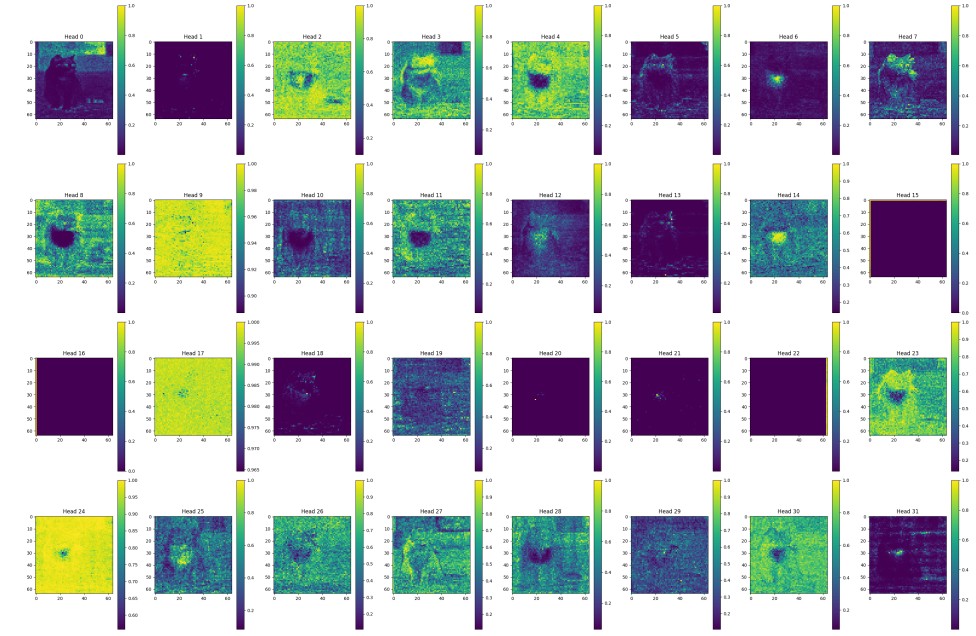

Figure 25: Visualization of the attention maps $\bar{A}_{ZP_2}^j$ across all attention heads, corresponding to prompt $P_2$ for the cat image in Fig. 3.

### A.6 LIMITATIONS AND BROADER IMPACT

**Limitations** The Instruction Influence Disentanglement (IID) framework, while effective, faces several constraints. It introduces computational overhead when processing multiple instructions and may struggle with highly complex or semantically overlapping instructions where influence regions cannot be clearly separated. Although we argue each instruction inside a given editing session is

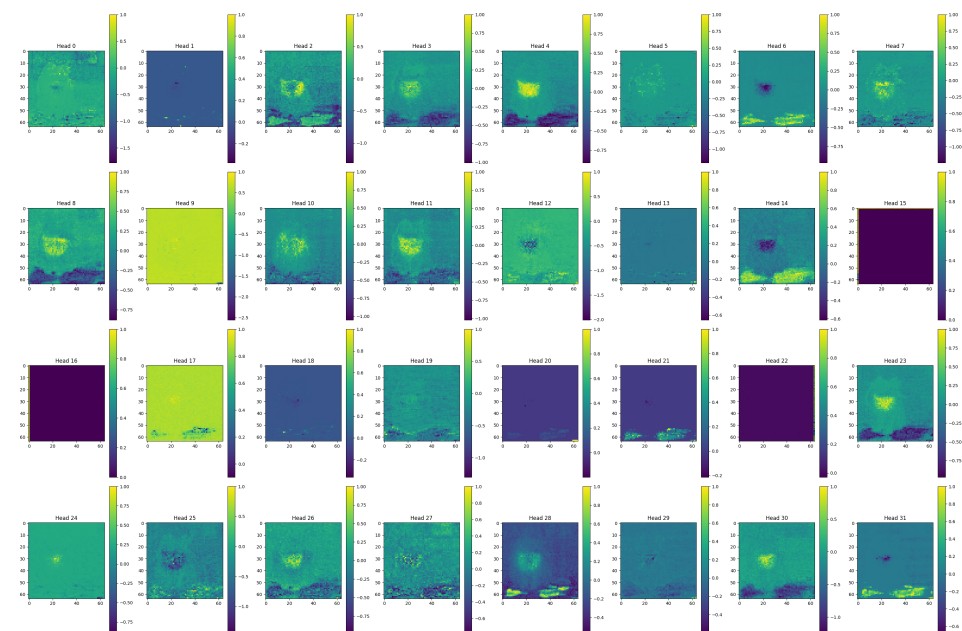

Figure 26: Visualization of the difference between $\bar{A}_{ZP_0}^j$ and $\bar{A}_{ZP_1}^j$ across all attention heads.

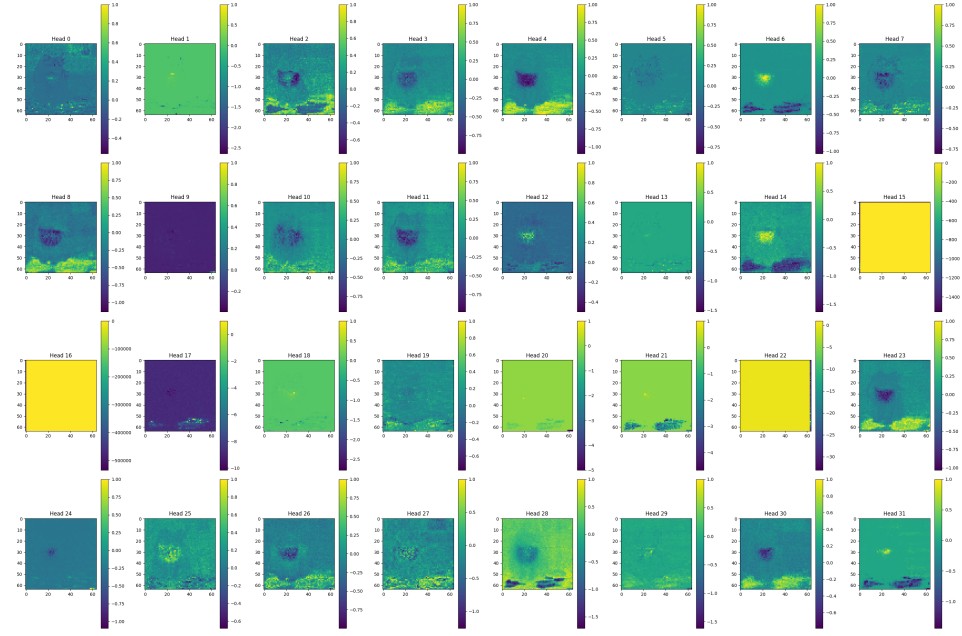

Figure 27: Visualization of the difference between $\bar{A}_{ZP_1}^j$ and $\bar{A}_{ZP_0}^j$ across all attention heads.

expected to focus on a distinct editing target, we further propose an automatic schedule strategy to deal with such mask conflicts in Appendix. A.3. Additionally, our framework is specifically designed for DiT-based architectures, restricting its direct application to other diffusion model variants without significant modifications.

**Broader Impact.** IID has the potential to democratize image editing by making complex manipulations more accessible to non-experts through natural language instructions, while also improving efficiency through reduced diffusion steps that lower computational requirements. However, like

all image manipulation technologies, it presents dual-use concerns and could be misused to create misleading content. On the positive side, this technology could transform workflows in creative industries by enabling faster iteration and more nuanced control, while making advanced editing more accessible to people with varying technical backgrounds. The insights gained about attention mechanisms in DiT models may also inform future research beyond image editing in multimodal AI systems.

### A.7 LLM USAGE

In this paper, we primarily utilize large language models for language refinement, encompassing grammar correction and stylistic enhancements. Additionally, LLMs are employed to assist in verifying mathematical proofs.

