# OpenReview forum: "Disentangling Instruction Influence in Diffusion Transformers for Parallel Multi-Instruction-Guided Image Editing"
_ICLR.cc/2026/Conference — ICLR 2026 Conference Withdrawn Submission_

### Official Review · Reviewer_5pFe · 2025-10-28

[review text omitted: it was posted to a different submission]

---

> ### Author Response · Authors · 2025-11-22
> **Part 1 to Reviewer 5pFe**
>
> We appreciate you for reading our paper carefully and providing insightful reviews.  Our point-to-point response is listed as follows:
>
>
>
> **Weakness 1 and Question 1**
>
> Thank you for pointing out this error. As suggested by the phrase "the negative values are then set to zero to suppress non-relevant regions," the correct operation should be a ReLU or max function. We have corrected this in the revised version of our paper (Eq. 3). We appreciate your careful reading and helpful feedback.
>
> **Weakness 2**
>
> We apologize for the confusion. Since both OmniGen and FluxKontext use positional embeddings, we pad all instructions to the same length using padding tokens and assign the same position ID sequence for each instruction before feeding them into the text encoder.
>
> For example, given two instructions: $P_1$ ("add an apple on the desk") with a token length of 6, and $P_2$ ("remove leaves") with a token length of 2, we pad $P_2$ to a length of 6. Consequently, both instructions share the same position ID sequence, such as  $[1, 2, 3, 4, 5, 6]$. We have added this explanation in lines 294–295 of the revised version.
>
>
> **Weakness 3 and Question 2**
>
> We are sorry for the confusion.  It is a hard mask where attention scores are set to $-\infty$. Given a squence of instruction $\{P_1,\ldots, P_N\}$, to make the token of $P_i$ can only attend to noisy image tokens excluding regions masked by $M_j$, where $j \in [1, N]$ and $j \neq i$, we set the attention scores between the token of $P_i$ and the excluded image regions to $-\infty$. In particular, when there are overlapping regions between $M_i$ and other masks, the $P_i$ tokens are still allowed to attend to the corresponding image tokens.
>
> We apologize for the confusion and have revised our paper accordingly.
>
>
> **Weakness 4**
>
> Thank you for this question. We agree with your opinion that the content in Appendix A.4 presents a theoretical rather than a principled or formally rigorous analysis. Specifically, Appendix A.4.2 provides a high-level, intuitive explanation of "instruction conflict" based on data sparsity and velocity field misalignment, rather than a fully principled derivation. A complete mathematical formulation is beyond our current capability and may be explored in future research.
>
> Nevertheless, this theoretical analysis motivates two potential solutions to address the challenges of multi-instruction image editing:
> (1) our IID framework, which isolates instructions using masks during diffusion, and (2) generating high-quality, complex instruction–image pairs, for which IID itself can serve as a data generator to enhance the editing model.
>
> The analysis also helps explain observed phenomena. For instance, when instruction masks overlap, the isolation strategy becomes less effective due to unintended attention across conflicting regions. In such cases, the model’s inherent capacity becomes the dominant factor for success, and simple averaging strategies tend to offer limited benefit. This is further discussed and empirically supported in Appendix A.2.6 of the revised version. Therefore, we believe that even non-rigorous theoretical insights can offer valuable intuition.
>
> Importantly, we would like to emphasize that theoretical contributions are not the main focus of our work. As stated in the introduction, the key contributions of our paper are as follows:
>
> 1) We conduct an in-depth investigation of self-attention mechanisms in DiTs for instruction-guided image editing, uncovering unexplored insights to inform future research.
>
> 2. We propose a novel framework that enables the parallel processing of multiple edits in a single denoising process. It not only significantly reduces diffusion steps but also improves editing performance, including lower noise generation and better consistency in non-edited regions compared to step-by-step editing.
>
> 3. Our framework is extensively evaluated on open-source and custom-constructed multi-instruction editing datasets, demonstrating versatility on Omnigen, FluxEdit, and the recent state-of-the-art FluxKontext.

---

> > ### Author Response · Authors · 2025-11-22
> > **Part 2 to Reviewer 5pFe**
> >
> > **Weakness 5**
> >
> > Thank you very much for this question. In our original version, we compared our head-wise mask generation strategy with FOI’s mask extraction method (Object-Avg) in Fig. 7. The results demonstrate that FOI’s strategy fails to accurately identify the correct masks when applied to DiT-based models. Since accurate masks are essential for both FOI and our method, we speculate that FOI performs poorly on DiT architectures due to this limitation.
> >
> > To further support this hypothesis, we adapted FOI from U-Net-based diffusion models to DiT and evaluated its performance on the MagicBrush dataset (Omnigen), as shown in Table 1 in our revised version. The results show that IID significantly outperforms FOI. A qualitative comparison is also provided in Fig. 5, where FOI frequently fails to complete all given instructions and generates unrealistic objects, such as windows, in the final column.
> >
> > **This discussion has been added to Appendix A.2.9 in the revised version. We encourage you to refer to that section for more details.**
> >
> > **Weakness 6**
> >
> > Thank you for your good question. For multi-instruction image editing, sequential editing often leads to cumulative distortions and quality degradation, while naive concatenation of instructions frequently fails to fulfill all intended modifications due to instruction conflicts. To address these challenges, we propose the IID framework, which aims to disentangle the influence of multiple instructions and complete all instructions in a single editing turn.
> >
> > IID consists of three essential components, involving Head-wise Mask Generation, Adaptive Blender, and Multi-instruction Influence Disentanglement. The first component generates accurate masks for each instruction. The second combines the text tokens and noisy image tokens of all instructions into a compositional representation, enabling parallel execution of instructions in a single editing turn. The third component disentangles the influence of each instruction, thereby mitigating instruction conflicts and ensuring the completion of all intended notifications. **The output of each component feeds into the next, and all are necessary to achieve effective multi-instruction editing**.
> >
> > While various mask generation strategies and settings have been investigated in Sec. 5.3, Ap-
> > pendix A.2.4, and Appendix A.2.8, we focus here on evaluating the contributions of Adaptive Blender and Multi-instruction Influence Disentanglement, as shown in Fig. 22. . Since Adaptive Blender is
> > critical for enabling parallel editing, we assess its role by replacing the mask-based blending strategy
> > in Eq (5) with simple averaging, denoted as w/o Adaptive Blender, while keeping the remaining
> > components unchanged. Similarly, we evaluate the effect of removing the Influence Disentanglement
> > component by allowing each instruction to attend to the entire image. As illustrated in Fig. 22,
> > replacing Adaptive Blender with averaging leads to ambiguous edits, where visual attributes intended
> > for one region spill into others. For example, the clock inherits the color meant for the fork in the first row
> > of Fig. 22. Without the Influence Disentanglement module, instruction conflicts emerge, preventing
> > the model from fulfilling all instructed edits
> >
> > **This discussion has been appended to the Appendix. A.2.10 in our revision version.  We encourage you to refer to that section for more details.**
> >
> >
> >
> > **Weakness 7 and Question 5**
> >
> > Thank you for this question. In multi-instruction image editing, given a sequence of instructions $\{P_1, \ldots, P_N\}$, if the base model (Omnigen, FluxKontext) fails to execute a particular instruction in isolation, it is also unlikely to complete it in the multi-instruction setting. This applies not only to our proposed IID framework but also to other baseline methods such as Step-by-Step editing, NaiveCon, and FOI.
> >
> > During the construction of our own dataset, we filtered out only those individual instructions that were clearly beyond the capability of the base model, rather than discarding the entire instruction sequence. Therefore, we argue this will not introduce any bias that would artificially inflate the performance of our IID framework, as the same limitations also apply to the baseline methods.
> >
> > To ensure the robustness and generality of our conclusions, we further conducted experiments on the publicly available MagicBrush dataset. These additional evaluations support the reliability and consistency of our findings.

---

> > > ### Author Response · Authors · 2025-11-22
> > > **Part 3 to Reviewer 5pFe**
> > >
> > > **Question 3**
> > >
> > > Thank you for the question. As described in the Adaptive Blender section of Sec. 4.2, for each instruction, we update its corresponding masked region in the averaged representation using the respective latent feature $z_{S,i}$. For regions where masks from multiple instructions overlap, we replace them with the average of the corresponding $z_{S,i}$  to incorporate information from all relevant instructions (see lines 305–306).
> > >
> > > As a result, regardless of the blending order in Eq. (5), the final representation $\bar{z}_S$ remains mathematically identical. Consequently, the editing results are invariant to the order of blending. We have also verified this empirically and observed no variation in the outputs.
> > >
> > > **Question 4**
> > >
> > > Thank you for this question. As discussed in Appendix A.4.1 and illustrated in Fig. 3 ("Avg" column), the attention maps of DiT models exhibit diverse behaviors, i.e., some heads emphasize the edited region, others focus on unedited areas, or prioritize overall image reconstruction. This diversity suggests that simple head averaging is suboptimal for accurately identifying editing regions in DiTs. For example, in subfigure (c) of Fig. 3, the highest attention values in the "Avg" column do not correspond to the intended editing region, resulting in an incorrect mask.
> > >
> > > Note that Our Avg baseline has applied the same binarization method as IID (Otsu’s filter) after averaging attention maps across all heads, as described in Sec. 5.3.  However, as shown in Fig. 23  in our revised version, even emoving overlapping regions among masks, this approach still fails to produce accurate masks, further demonstrate the advantages of our proposed head-wise mask generation strategy.
> > >
> > > We hope we have addressed your concerns. If you have any concerns, please feel free to ask us.

---

> > > > ### Comment · Reviewer_5pFe · 2025-11-25
> > > >
> > > > The authors have addressed most of my concerns. I will raise my score and support the acceptance of this paper.

---

> ### Author Response · Authors · 2025-11-25
>
> Dear Reviewer 5pFe,
>
> Thank you very much for your thoughtful feedback! We are pleased to have addressed most of your concerns and will further revise our paper to incorporate all necessary improvements accordingly.
>
> We sincerely appreciate your time and your decision to improve your score!
>
> Best regards,
>
> The Authors

---

> ### Author Response · Authors · 2025-11-29
> **Corrected Review from Reviewer 5pFe (Part 1)**
>
> Dear AC and SACs,
>
> We would like to bring to your attention that Reviewer 5pFe initially submitted an incorrect review prior to the discussion phase. After we reported this to the last Area Chair, the reviewer was asked to revise and correct the review accordingly. However, during the rollback process, the review reverted to the earlier incorrect version.
>
> **Originally, the reviewer assigned a score of 6, but upon realizing the mistake, revised the score to 4. During the discussion phase, after acknowledging that most of his/her concerns had been addressed, the reviewer updated the score back to 6**.
>
> For your reference, we have attached the correct version of Reviewer 5pFe’s review.
>
>
> # Reviews of  Reviewer 5pFe
>
>
> **Weaknesses:**
>
> - Equation (3) which defines the core head-wise mask generation strategy, is mathematically incorrect. This expression will always yield a non-positive result, which is contrary to the goal of creating a positive mask over the target edit region. The text states, "The negative values are then set to zero to suppress non-relevant regions," which implies the operation should be a ReLU or `max(0, ...)` function. This is not a minor typo; it is an error in the formal definition of the central technical contribution of the paper and reflects a lack of rigor.
> - In Adaptive Blender (Sec 4.2), the sequential blending process described in Equation (5) is order-dependent. While the paper mentions re-ranking instructions for FluxEdit based on an "Influence Score" to mitigate this, it's unclear how this is handled for OmniGen and FluxKontext, for which the paper claims "we ensure all share the same position embedding, neutralizing positional bias." This is a vague statement. Positional embeddings typically relate to token sequence order, so it's not obvious how making them identical neutralizes the bias from the *blending* order in Eq. (6).
> - In Multi-instruction Influence Disentanglement (Sec 4.2), the description of the final attention masking step is highly ambiguous. The paper states that "the token of can only attend to noisy image tokens excluding regions masked by , where and ." How is this "gating" of attention implemented? Is it a hard mask where attention scores are set to ? Is it a soft modulation? This critical detail of the method is completely absent, making the mechanism opaque.
> - The theoretical analysis in Appendix A.4, while presented with formal notation, offers little genuine insight. Section A.4.1 merely recasts standard ODE numerical error analysis to argue that error accumulates over multiple steps, a widely known phenomenon. Section A.4.2 provides a high-level, intuitive argument for "instruction conflict" based on data sparsity and misalignment of velocity fields, but it lacks any rigorous derivation and does not constitute a meaningful theoretical contribution. This section feels more like a post-hoc justification than a principled analysis.
> - The empirical evaluation relies almost exclusively on two simple baselines: step-by-step (Step) and naive concatenation (NaiveCon). While the authors correctly note the scarcity of multi-instruction methods for DiTs, this does not excuse the lack of a more challenging comparison. No attempt was made to adapt principles from U-Net-based multi-edit methods (e.g., attention modulation from FOI) to a DiT architecture, even as a rudimentary baseline. Dismissing such approaches by citing "unresolved challenges" without elaboration is insufficient. The strong performance of IID is less impactful when only compared against methods that are known a priori to be flawed.
> - The ablation studies, while useful, do not fully dissect the contribution of each component of the IID pipeline. For example, what is the performance if only the adaptive blending is used without the subsequent attention masking? Or vice-versa? The method is a sequence of several heuristics (head-wise subtraction, latent blending, final attention masking), and their relative importance is not established. This makes it difficult to ascertain which part of the framework is truly responsible for the performance gains.
> - The paper curates a new dataset for evaluation. However, the filtering process described in Appendix A.1 ("human experts manually assess the quality of the resulting outputs. If the overall error rate exceeds 60%, the case is considered beyond the model's ability and is excluded") is a potential source of significant selection bias. This process may inadvertently filter out precisely the challenging cases where instruction conflicts are most severe, thereby inflating the apparent performance of the proposed method.

---

> > ### Author Response · Authors · 2025-11-29
> > **Corrected Review from Reviewer 5pFe (Part 2)**
> >
> > **Questions:**
> >
> > - Could the authors please clarify and correct the formulation in Equation (5)? Is the intended operation `max(0, ...)`?
> > - Can the authors provide a more precise explanation of how the "Multi-instruction Influence Disentanglement" step is implemented? Specifically, how do you enforce that tokens for instruction only attend to image regions not masked by other instructions ?
> > - Regarding the Adaptive Blender in Equation (6), what is the justification for claiming that shared position embeddings for OmniGen/FluxKontext "neutralize positional bias" when the blending operation itself is sequential and thus order-dependent? Have you experimented with different blending orders for these models?
> > - The ablation study compares your head-wise subtraction to an "Avg" method. Have the authors considered a simpler but potentially stronger baseline for mask generation, such as applying a threshold (e.g., Otsu's method) directly to the attention map for each instruction individually, and then resolving overlaps? This would isolate the benefit of the subtraction component more directly.
> > - Given the filtering process for your custom dataset, how can you be sure that you are not systematically removing the most difficult multi-instruction cases, which could inflate the reported performance of your method?

---

### Official Review · Reviewer_QEsG · 2025-10-31

**Soundness:** 3
**Presentation:** 3
**Contribution:** 3
**Rating:** 6
**Confidence:** 4

**Summary:**

The paper proposes Instruction Influence Disentanglement (IID), a training-free framework for parallel multi-instruction image editing within a single denoising pass. It targets Diffusion Transformer (DiT) models like FluxEdit, Omnigen, and FluxKontext, aiming to solve two main problems: cumulative errors from step-by-step editing and conflicts from naive instruction concatenation. IID analyzes self-attention patterns in DiTs and introduces a head-wise mask generation strategy. For each instruction, it subtracts the mean attention map of other instructions to isolate the editing region, then aggregates and binarizes these masks. Next, an adaptive blender composes instruction tokens and latent images at a predefined timestep S. It uses a re-ranking process with an influence score to mitigate any dominance effects. Finally, an attention mask constrains instruction tokens to their respective regions, enabling disentangled parallel edits. Experiments on MagicBrush and a custom dataset show consistent improvements over baselines across metrics like L1/L2, CLIP-I/T, and DINO, as well as in human preferences.

**Strengths:**

1. Proposes a novel, training-free parallel editing framework. It utilizes "instruction-wise attention subtraction" to leverage the model's own attention for disentangling multiple instructions, eliminating the need for external segmenters.
2. The evaluation is comprehensive, demonstrating consistent performance improvements across multiple mainstream DiT backbones (e.g., FluxEdit, Omnigen) and datasets.
3. Addresses a critical, real-world need in multi-instruction DiT editing, solving a key problem where existing methods fail or are ineffective.

**Weaknesses:**

1. While the paper provides a theoretical discussion of error accumulation and instruction conflicts in the appendix, the main text adopts a simple averaging scheme for overlapping regions during blending, without analyzing its potential to reintroduce conflicts or degrade boundary consistency.
2. The evaluation is primarily conducted on relatively simple editing tasks. The custom dataset construction using GPT-4o may introduce biases, and the human evaluation involves only 5 participants, which may not be sufficient for robust conclusions. The paper lacks comparison with more recent multi-instruction editing methods or adaptation of existing approaches to DiT architectures.
3. The paper does not provide sufficient analysis of scenarios where the method fails or performs poorly.

**Questions:**

1. The IS score used to guide instruction position re-ranking is an innovative idea. How stable is IS under paraphrasing, i.e., different phrasings of the same instruction?
2. In applying masks within the DiT denoising process, how does IID compare against using minimally adapted external segmenters (e.g., SAM) to derive instruction-specific masks for DiTs? Does an external mask improve completion or fidelity, or introduce new artifacts and complexity?

---

> ### Author Response · Authors · 2025-11-22
> **Part 1 to Reviewer QEsG**
>
> Thank you very much for your insightful and positive scores. Our point-to-point response is listed as follows:
>
>
>
> **Weakness 1**
>
>
>
> Thank you for this insightful question. As explained in Appendix A.3  and the implementation details in Appendix A.1 in our original version, **we hypothesize that, in typical multi-instruction parallel editing scenarios, each instruction ideally targets a distinct image region with minimal or no mask overlap**.
>
>
>
> Moreover, overlapping regions between masks of different instructions represent a major source of failure, as illustrated in Fig. 18 and discussed in Appendix A.2.6 of the revised version. Based on the theoretical analysis in Appendix A.4, the model’s intrinsic capacity will become the key determinant of editing success in such cases. Stronger models are more likely to succeed. This is empirically validated in Fig. 19, where FluxKontext successfully completes both conflicting instructions, while OmniGen completes only one.
>
>
>
> This phenomenon is arguably expected. As analyzed in Appendix A.4.2, instruction conflicts may arise from velocity misalignment during the diffusion process and the lack of large-scale multi-instruction training data. Two potential solutions are: (1) our IID framework, which isolates instructions using masks during diffusion, and (2) generating high-quality, complex instruction–image pairs, for which IID itself can serve as a data generator to enhance the editing model. However, when instruction masks overlap, the first strategy becomes ineffective, as instructions may still attend to conflicting regions. In such cases, the model's capacity becomes the dominant factor, and averaging strategies offer limited improvement.
>
>
>
> To improve IID's generality under overlapping conditions, we introduced an Optimal Instruction Grouping strategy in Appendix A.3 of our original version. This method partitions instructions into non-conflicting groups while minimizing the number of editing turns required. We encourage readers to refer to Appendix A.3 for further details.
>
>
>
> **All of the above discussions have been incorporated into Appendix A.2.6 of our revised version.**
>
>
>
> **Weakness 2**
>
>
>
> **(1) Evaluation is primarily conducted on relatively simple editing tasks**
>
>
>
> In our work, due to the scarcity of multi-instruction image editing datasets, we primarily conduct experiments on the MagicBrush dataset and our own constructed dataset. For our custom dataset, image data and partial instructions are sourced from two open-source datasets, including EmuEdit [3] and OmniEdit  [4], as well as a proprietary dataset designed for real-world scenarios (excluding any non-publicly releasable content). The three open-source datasets (MagicBrush, EmuEdit, OmniEdit) have been widely used in the evaluation of image editing tasks. For example, one of the multi-objective editing methods, ParaEdit, conducted experiments on  MagicBrush, and Omigen conducted experiments on  EmuEdit [3].  The combined test samples cover a wide range of editing tasks, including object addition, removal, and style transformation, making the benchmark diverse and representative of real-world use cases.
>
>
>
> Additionally, during the rebuttal phase, we conducted supplementary experiments on the multi-instruction Kontext-Bench [4], which lacks ground-truth images and thus requires human evaluation. As shown in Figs. 16 and 17 of our revised version, our proposed IID framework outperforms both Step and NaiveCon in terms of Instruction Alignment and Image Alignment.
>
>
>
> **(2) Bias by GPT-4o**
>
>
>
> Thank you for this question. Although GPT-4o is used to generate editing instructions, we improve instruction quality through a two-stage process. First, we prompt GPT-4o to generate a rationale for each instruction and use this to filter out illogical or ambiguous instructions. Second, all instructions are processed by the editing model, and the resulting outputs are manually evaluated by human experts to ensure data quality.  Therefore, we argue that there is no significant bias in the instruction set. These details are further discussed in the Human Preference Study Section of Appendix A.1.

---

> > ### Author Response · Authors · 2025-11-22
> > **Part 2 to Reviewer QEsG**
> >
> > **(3) Human Evaluation**
> >
> > Thank you very much for pointing that out.   In the revised paper, we expanded our user study to include 40 participants and updated the human preference results, as shown in Fig. 6 of the revised version. Across different datasets and backbone editing models, the results consistently indicate that as the number of editing instructions increases, IID achieves steadily higher preference rates in both Instruction Alignment and Image Alignment. In contrast, the performance of Step and NaiveCon consistently declines under the same conditions.
> >
> > We also report statistical significance using the Exact Binomial Test in Fig. 16 of the Appendix of the revised version. The analysis shows that IID significantly outperforms NaiveCon in Instruction Alignment, and with increasing instruction counts, also significantly surpasses Step. Similarly, for Image Alignment, IID consistently outperforms Step, with win rates of approximately 70%–80%, and also exceeds the performance of NaiveCon.
> >
> >
> >
> > **For further details, please refer to the Human Preference Study in Sec. 5.2 and the Supplementary Experiments of Human Study in Appendix A.2.5. of our revised version.**
> >
> > **(4) More recent multi-instruction editing methods**
> >
> > Thank you for this question. We compared with ParaEdit [2] in Table 1 of our original version and compared our head-wise mask extraction strategy with FoI's [1] mask extraction strategy in Fig. 7.  As suggested by other reviewers, we adapted FOI [1] to DiT-based methods in Table 1 and revised Fig. 5 in our revised version. These results further highlight the advantages of our IID framework.
> >
> > **Weakness 3**
> >
> > Thank you for this insightful question. We have **added a failure case analysis of our proposed IID in Appendix A.2.6 of the revised version**. We encourage you to refer to that section for full details. Below, we briefly summarize the key findings:
> >
> > We identify two primary sources of failure. The first involves conflicts in editing regions among multiple instructions. As shown in the second row of Fig. 18, overlapping areas between the headband and the laptop lead to the failure of the instruction to remove the headband. Similarly, in the third row, the overlap between the flower and the wooden board results in the failure to add the flower.
> >
> > The second type of failure occurs when individual instructions exceed the model’s editing capacity. For example, as shown in the first row of Fig. 18, the instruction to remove the milk box cannot be fulfilled by OmniGen, even when executed independently. As a result, the model also fails to complete this instruction when multiple edits are applied.
> >
> > **Question 1**
> >
> > Thank you for this question!  In the Adaptive-Blender component of IID, we propose computing the Influence Score (IS) for each instruction to quantify its relative impact on the editing process. The instructions are then sorted in ascending order based on their IS values to approximately equalize their influence during editing.
> >
> > To examine how the paraphrasing of instructions affects the scale of IS, we present a qualitative example in Fig. 20. The results show that **rephrasing an instruction can lead to differences in its IS**; however, **this does not affect the final editing result**. For example, in the right of Fig. 20, the IS of the first instruction increases compared to the middle one, but the re-ranking strategy ensures that all instructions are still executed appropriately.
> >
> > We have added this discussion in Appendix A. 2.7 in our revised version.
> >
> > **Question 2**
> >
> > Thank you for this question! We compare our mask extraction strategy with a SAM-based pipeline, which first employs Grounding DINO to identify the bounding box of the given target object in the editing instructions, followed by SAM for semantic segmentation to generate the corresponding mask, as shown in Fig. 21 of our revised version. For relatively simple editing tasks such as object removal, this approach is capable of producing high-quality masks. **However, the SAM-based pipeline requires explicit specification of the object to be segmented and thus struggles with instructions involving non-existent elements, such as object addition. Moreover, it is limited in handling background modifications and cannot accurately interpret complex instructions, such as spatial constraints or specific quantities.
> >
> > We have added this discussion to Appendix A. 2.8 in our revised version.
> >
> > We sincerely thank you again for your patience and insightful reviews.  If you have other concerns, please feel free to aks us.

---

> > > ### Author Response · Authors · 2025-11-22
> > > **Reference to Reviewer QEsG**
> > >
> > > [1] Guo, Qin, and Tianwei Lin. "Focus on your instruction: Fine-grained and multi-instruction image editing by attention modulation." Proceedings of the IEEE/CVF Conference on Computer Vision and Pattern Recognition. 2024.
> > >
> > > [2] Huang, Mingzhen, et al. "ParallelEdits: Efficient Multi-Aspect Text-Driven Image Editing with Attention Grouping." Advances in Neural Information Processing Systems 37 (2024): 22569-22595
> > >
> > > [3] Sheynin, Shelly, et al. "Emu edit: Precise image editing via recognition and generation tasks." *Proceedings of the IEEE/CVF Conference on Computer Vision and Pattern Recognition*. 2024.
> > >
> > > [4] Wei, Cong, et al. "Omniedit: Building image editing generalist models through specialist supervision." *The Thirteenth International Conference on Learning Representations*. 2024.
> > >
> > > [4] Batifol, Stephen, et al. "FLUX. 1 Kontext: Flow Matching for In-Context Image Generation and Editing in Latent Space." **arXiv e-prints** (2025): arXiv-2506..

---

> > > > ### Comment · Reviewer_QEsG · 2025-11-26
> > > >
> > > > Thanks for your detailed response. I'll maintain my positive rating.

---

> > > > > ### Author Response · Authors · 2025-11-26
> > > > >
> > > > > Dear Reviewer QEsG,
> > > > >
> > > > > Thank you very much for your constructive feedback, and thank you again for your positive scores!
> > > > >
> > > > > Best Regards!
> > > > >
> > > > > Authors

---

### Official Review · Reviewer_oZHh · 2025-10-31

**Soundness:** 3
**Presentation:** 3
**Contribution:** 2
**Rating:** 4
**Confidence:** 3

**Summary:**

The paper addresses multi-instruction-guided image editing in DiTs, where users specify several natural-language instructions simultaneously. Existing strategies tend to degrade image quality or yield incomplete edits. The authors overcome these limitations via introducing IID, a training-free framework enabling multiple edits within a single denoising process. It constructs head-wise attention masks to blend instruction-specific latents, and builds a compositional attention mask that ensures local edits while preserving non-edited regions. Experiments demonstrate that IID improves image fidelity and efficiency across Omnigen, FluxEdit, and FluxKontext models.

**Strengths:**

1.The introduction precisely identifies two failure cases in multi-instruction editing—error accumulation in sequential steps and instruction conflicts in prompt concatenation. This evidence grounds the paper’s motivation in observable limitations of prior methods.

2.The proposed mask derivation introduces a simple yet effective subtractive strategy across attention heads to isolate editing regions.

3.The adaptive blender and re-ranking strategy mitigate instruction dominance, ensuring balanced edits.

4.Good performance achieved under both MagicBrush and a custom dataset extending to six-instruction cases.

**Weaknesses:**

1. The adaptation to the proposed method is limited to some extent. According to the experiments from the paper, the method only works in DiT-based architectures.

2.One of my concerns is the actual application. Compared with the parallel instructions paradigm, sequential instruction editing is usually used in real-world application as the user can adjust the requirement at the next time. However, the parallel instruction is equivalent to the T2I editing that the text is pointwise and concise, and has differences compared with sequential image editing.

3. Based on the stated concern, it is better to show the generation performance with the traditional generated model with the same type of instructions to demonstrate the effectiveness.

4. Human evaluation involves only five participants, lacking statistical analysis or inter-rater reliability, which undermines the reproducibility of subjective results.

5.The subsequent thresholding with Gaussian + Otsu filters introduces multiple heuristic stages. How about the performance relative to other mask computation settings?

**Questions:**

Please see the weakness above.

---

> ### Author Response · Authors · 2025-11-22
> **Part 1 to Reviewer oZHh**
>
> Thank you very much for your insightful reviews, which have helped me improve the quality of the paper a lot.  Our point-to-point response is listed as follows:
>
> **Weakness 1**
>
> Thank you for your question. We agree with your opinion. However, our work focuses on DiT-based architectures because **DiT-based models have demonstrated significantly superior performance in image editing compared to all U-Net-based methods**, such as SD-SDEdit [4], Null Text Inversion (NTI) [5], ParaEdit [2], HIVE, InstructPix2Pix, and others, as shown in the introduction and the experimental results in Table 1. Moreover, recent state-of-the-art open-source models, including OmniGen, FluxKontext, and Qwen-Image, are all built on the DiT architecture. Based on these trends, our proposed IID framework is specifically designed to address the challenges of parallel multi-instruction image editing in the DiT-based paradigm.
>
>
>
> **Weakness 2**
>
> Thank you for your good question! First, while the potential user may need step-by-step editing to iteratively refine instructions, **such a sequential approach has significant limitations**, including progressive distortion and quality degradation. This often leads to suboptimal results, as shown in Fig. 1, Fig. 4, Table 1, and Fig. 6 in our original work. These issues persist even in state-of-the-art DiT-based models. In practice, users can iteratively refine their instructions and apply our proposed IID framework with the finalized instruction set, resulting in more stable and higher-quality outputs.
>
>
>
> Second, **there exist numerous real-world scenarios requiring multi-instruction editing, especially for business applications in the real world**. For example, one-click enhancement features in photo editing apps, large-scale e-commerce platforms generating diverse color or background variations for the same product [3], require balancing both editing efficiency and image quality.  It is worth noting that IID supports parallel execution of multiple instructions in a single diffusion process, offering substantial efficiency gains. **Specifically, IID can achieve up to a 4× speedup over Omnigen and a 2× speedup over FluxKontext compared to sequential editing approaches, as shown in Table 5 of our original version.**
>
>
>
> Third, **IID provides a practical solution for generating more complex and high-quality image editing data for training image editing models** in a more efficient manner, whereas creating data through Photoshop is difficult to scale, and existing editing models struggle with such complex multi-instruction scenarios.
>
>
>
> Regarding text-to-image editing baselines, we reported the performance of SD-SDEdit [4], Null Text Inversion (NTI) [5], and ParaEdit [2], which are tailored for multi-objective editing, in Table 1 of our original version. However, DiT-based instruction-guided methods consistently outperform these caption-guided baselines, further demonstrating that our IID framework is better suited for complex, real-world multi-instruction editing scenarios.
>
>
>
> **Weakness 3**
>
>
>
> Thank you for your question. In the original version of our paper, we have compared IID with traditional generation models using the same instruction types, as shown in Table 1 and Fig. 5. For text-to-image (T2I) editing, referred to as target caption-guided baselines, we evaluated SD-SDEdit [4], Null Text Inversion (NTI) [5], and ParaEdit [2]. For instruction-guided Unet-based baselines, we included  HIVE, InstructPix2Pix (InP2P), HQ-EDIT, InP2P-Magic, and SmartEdit. Please refer to the Baseline subsection of the Sec. 5.1 and Appendix A.1 for further details. The results show that these traditional image editing models consistently underperform compared to DiT-based instruction-guided models such as Omnigen and FluxKontext.
>
>
>
> **If you have suggestions for additional baselines, we would be happy to include them in future experiments.**

---

> ### Author Response · Authors · 2025-11-22
> **Part 2 to Reviewer oZHh**
>
> **Weakness 4**
>
> Thank you for the valuable question. In the revised paper, we expanded our user study to include 40 participants and updated the human preference results, as shown in Fig. 6 of the revised version. Across different datasets and backbone editing models, the results consistently indicate that as the number of editing instructions increases, IID achieves steadily higher preference rates in both Instruction Alignment and Image Alignment. In contrast, the performance of Step and NaiveCon consistently declines under the same conditions.
>
> We also report statistical significance using the Exact Binomial Test in Fig. 16 of the Appendix of the revised version. The analysis shows that IID significantly outperforms NaiveCon in Instruction Alignment, and with increasing instruction counts, also significantly surpasses Step. Similarly, for Image Alignment, IID consistently outperforms Step, with win rates of approximately 70%–80%, and also exceeds the performance of NaiveCon.
>
> **For further details, please refer to the Human Preference Study in Sec. 5.2 and the Supplementary Experiments of Human Study in Appendix A.2.5. of our revised version.**
>
>
> **Weakness 5**
>
> Thank you for your this question.  It is worth noting that our core contribution lies in the head-wise mask extraction strategy (Eq. 3), designed for DiT-based diffusion models, motivated by the attention-based interpretability analysis in Sec. 4.1. While our default mask refinement pipeline, comprising a Gaussian filter followed by Otsu’s thresholding (Eq. 4)  has proven effective, these components can be replaced with alternative engineering strategies. In our approach, the Gaussian filter is applied to suppress high-frequency noise, ensuring a smoother and more coherent mask. Otsu’s filter then adaptively determines an optimal threshold that maximizes inter-class variance, allowing for effective separation of foreground and background without manual tuning.
>
>
>
> To investigate of different mask computation settings, we qualitatively compare several alternative configurations in Fig. 15 in our revised version: (a) Gaussian filter followed by multiple fixed thresholds for binarization (b) Otsu’s filter without Gaussian smoothing, (c) median filtering followed by Otsu’s filter, (d) Otsu’s filter with morphological operations (specifically,  erosion followed by dilation), and (e) Gaussian filter combined with Otsu’s filter and morphological operations.}
>
>
>
> Our findings are as follows. First, using fixed thresholds is highly sensitive to the specific test case, as the value ranges of attention maps vary significantly across instructions, even after Gaussian smoothing, thus rendering fixed thresholds unsuitable for general use. In contrast, Otsu’s filter effectively addresses this issue by providing adaptive thresholding, thus improving generalization. Second, morphological operations and median filtering can serve a similar role to Gaussian filtering by suppressing high-frequency noise, resulting in cleaner masks with reduced artifacts. Lastly, we find that the combination of Gaussian filter and Otsu’s thresholding already produces high-quality masks in most cases, often making additional morphological operations unnecessary.
>
>
>
> **We have included this discussion in Appendix A.2.4. in our revised version.**
>
>
>
> We hope we have addressed your cocnerns. If you have any concern, please feel free to ask us.
>
>
>
> [1] Guo, Qin, and Tianwei Lin. "Focus on your instruction: Fine-grained and multi-instruction image editing by attention modulation." Proceedings of the IEEE/CVF Conference on Computer Vision and Pattern Recognition. 2024.
>
>
>
> [2] Huang, Mingzhen, et al. "ParallelEdits: Efficient Multi-Aspect Text-Driven Image Editing with Attention Grouping." Advances in Neural Information Processing Systems 37 (2024): 22569-22595.
>
>
>
> [3]  https://pixlr.com/blog/automating-photo-edits-the-role-of-ai-in-batch-editing/
>
>
>
> [4] Shi, Yichun, Peng Wang, and Weilin Huang. "Seededit: Align image re-generation to image editing." arXiv preprint arXiv:2411.06686 (2024).
>
>
>
> [5] Mokady, Ron, et al. "Null-text inversion for editing real images using guided diffusion models." Proceedings of the IEEE/CVF conference on computer vision and pattern recognition. 2023.
>
>
>
> [6] Batifol, Stephen, et al. "FLUX. 1 Kontext: Flow Matching for In-Context Image Generation and Editing in Latent Space.arXiv e-prints (2025): arXiv-2506.
>
>
>
> [7] Hertz, Amir, et al. "Prompt-to-prompt image editing with cross attention control." arXiv preprint arXiv:2208.01626 (2022).

---

> ### Author Response · Authors · 2025-11-27
>
> Dear  Reviewer oZHh,
>
> Thank you very much for reviewing our paper again! As the discussion period is set to end in one week, we would like to inquire if you have any remaining concerns regarding our paper. We are happy to clarify any remaining questions or concerns.
>
> Best Regards!
>
> Authors

---

### Note · Authors · 2026-01-28

**Comment:**

Concern regarding AC comments:

Handling Overlapping and Global Instructions:

Our method supports edits involving overlapping regions and global instructions. **This limitation has been identified by Reviewer QEsG** and addressed during rebuttal. As illustrated in the second example of the right column in Fig. 1, IID inherently accommodates overlapping edits. We further address this with our proposed Optimal Instruction Grouping strategy (see Appendix A.3).

Realism of Examples and Benchmarks:

Our experiments are grounded in established open-source benchmarks (EmuEdit, OiniEdit, MagicBrush). Additionally, we have provided results on the newly released FluxKontext benchmark, which reflects realistic multi-instruction editing scenarios. What do AC mean, "actual use cases of sequential editing" and "complex example"?

Novelty and the actual algorithm is a simple region manipulation:

Our work is the first to explore **training-free multi-instruction** image editing for **DiT models**. Both the mask extraction strategy used in DIT and the Optimal Instruction Grouping algorithm are novel and have not appeared in prior literature.

**Withdrawal Confirmation:**

I have read and agree with the venue's withdrawal policy on behalf of myself and my co-authors.

---

### Meta-Review · Area_Chair_5iZG · 2025-12-24

**Summary:**

The reviewers provided the following concerns:
1) The human evaluation was only done with 5 participants
2) The provided examples do not use descriptions that are similar to actual use cases of sequential editing and are too simple
3) The actual algorithm is a simple region manipulation
4) Not enough analysis of failure cases
5) The new dataset may have significant selection bias and does not capture the range of possible multi-step edits

**Reviewer Concerns:**

I believe that points 1 and 4 were partially addressed in the rebuttal phase. The reviewers were overall slightly positive, but still borderline.

Overall, I think there are two main issues that are not really addressed by the reviewers and that additionally make this work problematic:
1) The method is a simple region manipulation method. This method may work in some cases, but not in others. There are many examples of multi-step instructions that affect overlapping regions or contain global instructions.
2) The time of simple region manipulation algorithms in image editing is a thing of the past. This is not a modern topic or an interesting method.

I just think the technical innovation is not sufficient for an ICLR paper.

**Reviewer Scores:**

I think the average score would have stayed the same with all reviewers keeping their score at 5.33. However, there are additional problems I highlighted in the summary, so overall I recommend rejection. The decision could be revised upwards.

---

### Decision · Program_Chairs · 2026-01-26

Reject